# Tigerfish designs oligonucleotide-based in situ hybridization probes targeting intervals of highly repetitive DNA at the scale of genomes

Robin Aguilar[1], Conor K. Camplisson[1], Qiaoyi Lin[1], Karen H. Miga [2,3], William S. Noble [1,4] & Brian J. Beliveau [1,5,6]

Fluorescent in situ hybridization (FISH) is a powerful method for the targeted visualization of nucleic acids in their native contexts. Recent technological advances have leveraged computationally designed oligonucleotide (oligo) probes to interrogate >100 distinct targets in the same sample, pushing the boundaries of FISH-based assays. However, even in the most highly multi-plexed experiments, repetitive DNA regions are typically not included as targets, as the computational design of specific probes against such regions presents significant technical challenges. Consequently, many open questions remain about the organization and function of highly repetitive sequences. Here, we introduce Tigerfish, a software tool for the genome-scale design of oligo probes against repetitive DNA intervals. We showcase Tigerfish by designing a panel of 24 interval-specific repeat probes specific to each of the 24 human chromosomes and imaging this panel on metaphase spreads and in interphase nuclei. Tigerfish extends the powerful toolkit of oligo-based FISH to highly repetitive DNA.

Fluorescent in situ hybridization (FISH) is a powerful technique that can reveal the spatial positioning and abundance of DNA and RNA molecules in fixed samples with subcellular resolution. Since their introduction in 1969[1], ISH and later FISH[2–4] methods have been refined to improve their detection efficiency and sensitivity[5]. One important technical development has been the introduction of synthetic DNA oligonucleotides (oligos) as a source of probe material[6]. Oligo-based probes offer important advantages over more traditional probes deriving from isolated genomic material, as oligo probes can be designed to have specific thermodynamic properties and programmed to contain stretches of exogenous sequences that can serve as 'readout' domains via the 'secondary' hybridization of a labeled,

complementary oligo. These advantages have led to the introduction of a growing set of 'spatial genomics' and 'spatial transcriptomics' methods that use complex 'probe sets' of many distinct oligo species[7–10] in combination with iterative rounds of secondary hybridization to visualize dozens or more genomic regions[11–14] and thousands or more RNA species[15–17], respectively, in the same cell or tissue sample.

The rapid adoption of oligo probes as a source of FISH probe material has also catalyzed the parallel development of computational tools for oligo probe design. These tools—which include OligoArray[18], PROBER[19], Chorus[20], mathFISH[21], OligoMiner[22], iFISH[23], ProbeDealer[24], Chorus2[25], and PaintSHOP[26]—aim to identify short windows of genomic sequence that have suitable thermodynamic and sequence properties

[1]Department of Genome Sciences, University of Washington, Seattle, WA, USA. [2]Department of Biomolecular Engineering, University of California Santa Cruz, Santa Cruz, CA, USA. [3]UC Santa Cruz Genomics Institute, University of California, Santa Cruz, CA, USA. [4]Paul G. Allen School of Computer Science and Engineering, University of Washington, Seattle, WA, USA. [5]Brotman Baty Institute for Precision Medicine, Seattle, WA, USA. [6]Institute for Stem Cell and Regenerative Medicine, University of Washington, Seattle, WA, USA. ✉e-mail: wnoble@uw.edu; beliveau@uw.edu

to serve as FISH probes. Once identified, 'candidate' probes are next screened for specificity to predict whether they will have off-target sites in addition to their intended target. This specificity screening typically relies on using alignment programs such as BLAST[27] or Bowtie2[28] to search for regions with high sequence similarity to the candidate probes, the use of k-mer counting programs such as Jellyfish[29] to assess whether the candidate probes contain k-mers (i.e., substrings) with high abundance in the genome of interest, or a combination of both approaches. After this specificity screening, candidate probes with predicted off-target binding are filtered and a final set of target-specific oligo probes is returned.

A key advantage of oligo probes is that they can be designed specifically to avoid targeting repetitive sequences. Repetitive sequences are frequent sources of unwanted background when performing in situ hybridization experiments due to their high copy number, and a set of "suppressive hybridization" methods using unlabeled repetitive DNA from the $C_0t$-1 fraction[30] as a blocking agent have been introduced to abrogate this background when using probes derived directly from genomic material[31–33]. Such blocking agents are generally not needed when using oligo probes, however, as computational oligo probe design methods either avoid discovering candidate probes in sequence annotated as being repetitive by tools like RepeatMasker[18–20,22,26,34] or purposefully filter candidate probes that align many times to the genome[18,20–26] or contain highly abundant k-mers[22,23,25,26]. As a result, while computational oligo probe design tools are able to operate at the scale of whole plant and mammalian genomes to produce repositories of tens of millions of oligo probes[23,26], a substantial fraction of large and complex genomes remains intentionally uncovered due to the presence of repetitive sequences.

Repetitive DNA accounts for ~50% of the human and mouse genomes and often even higher percentages in the genomes of plants[30,35,36]. Broadly, repetitive DNA falls into two categories: 1) Interspersed repeats such as SINE, LINE, and ALU elements that often occur as short, spatially isolated intervals within larger blocks of non-repetitive sequence[35]; 2) long tandem repeat arrays such as alpha satellite, human satellites 1–3, and the 45 S ribosomal DNA at which a single monomer is repeated many times to form multi-megabase intervals of repetitive sequence that are frequently located in pericentromeric regions and on the short arms of acrocentric chromosomes[36,37]. Collectively, repetitive DNA sequences are central to a set of diverse and essential cellular and organismal functions, including the recruitment of the chromosome segregation machinery during mitosis, the encoding of essential information such as the 47 S rRNA[38] and the replication-dependent histone genes[39], and the protection of chromosome ends[40]. Moreover, repetitive sequences are an important source of novel genic and regulatory sequences[41] and are hypothesized to be actively involved in potent evolutionary processes such as meiotic drive and speciation[42]. Thus, more detailed studies of highly repetitive DNA regions and their transcription products through low-cost targeted assays such as FISH may help uncover the mechanisms by which these mysterious regions exert their influence on important biological processes. For instance, the targeted visualization of repetitive regions would allow the assessment of chromatin compaction at the single cell level[43], the quantification of mitotic errors such as anaphase bridges[44], and the investigation of the micronucleation frequency of a given element[45]; as repetitive DNA regions are frequent sources of mitotic errors[46], such experiments may help define the mechanisms by which genome stability is maintained.

When desired, repetitive intervals make highly robust and effective FISH targets, as one or a few probe species can bind many times and thus produce a very large, bright signal at low cost. Indeed, all of the initial ISH targets were repetitive[1,47], and repetitive targets continue to be used routinely for diagnostic assays such as aneuploidy detection via interphase chromosome enumeration[48]. However, the deployment of probes against repetitive targets either requires the isolation and

experimental validation of cloned genomic material or a priori knowledge of experimentally validated oligo sequences. Computational approaches have been introduced to identify tandem repeat regions in worm[49] and plant systems[50,51] to select candidate chromosome-specific imaging oligo probes for experimental validation. However, neither these approaches nor computational tools designed to target non-repetitive regions provide a computationally scalable way to assess the predicted in situ behavior of oligo probes targeting repetitive DNA in the background of large and complex genomes.

Here, we introduce Tigerfish, a computational ecosystem tailored for the design and characterization of oligo probes targeting intervals of repetitive DNA at the genome scale. Tigerfish provides all functionality needed for discovering repetitive regions de novo, designing candidate probes, and performing deep in silico profiling of predicted binding activity. Tigerfish is open source, freely available, supported by extensive documentation and tutorials, and ships with a dedicated set of utilities to make it easier for users to visualize the predicted experimental outcomes of their designs. We showcase the utility of Tigerfish by designing and experimentally validating at least one interval-specific repeat probe for all 24 human chromosomes on metaphase spreads and augment these data by performing interphase enumeration of chromosomal copy number in human primary lymphocytes for all 24 human chromosomes. Finally, we provide a comprehensive catalog of probes and their predicted associated binding specificities that have been discovered by Tigerfish in the fully assembled human T2T CHM13v2 + HG002 chrY genome released by the Telomere-to-Telomere Consortium[36]. As our knowledge of the complete sequence of highly repetitive regions and how these regions vary amongst individuals and populations continues to increase from efforts such as the Human Pangenome Project[52] and Vertebrate Genomes Project[53], we anticipate that Tigerfish will play a key role in a number of applications including genome assembly variation, in situ karyotyping, and biological discovery.

## Results
### Challenges associated with designing probes that target repetitive DNA

We set out to design a computational pipeline optimized for the design of oligo-based in situ hybridization probes against intervals of repetitive DNA. Such intervals present unique design challenges. In order to frame these challenges, it is helpful to first consider the general case of probe design. In order to function effectively, in situ hybridization probes need to maximize important criteria: 1.) Efficacy—they must bind stably to their genomic or transcriptomic target and remain associated throughout the duration of hybridization and wash steps; 2.) Specificity—they must aggregate detectable (e.g., fluorescent) label at their intended target such that on-target signal rises above the levels of noise and off-target background labeling. While adjustments to the specified length and %G + C content ranges are sometimes required to accommodate repetitive intervals with skewed nucleotide compositions, in general the design of probes with high efficacy against repetitive DNA intervals is straightforward with any computational design program capable of performing nearest neighbor thermodynamic[54] calculations. Thus, if fed the proper input sequence, efficacious probes targeting repetitive DNA can be readily outputted by OligoArray[18], OligoMiner[22], iFISH[23], ProbeDealer[24], Chorus2[25], PaintSHOP[26], and other computational tools focused on oligo probe design. In contrast and as explained below, the evaluation of the specificity of probes targeting repetitive DNA intervals is a much more involved and challenging proposition.

Existing computational probe design methods attempt to avoid designing non-specific probes using one or more of the follow three main approaches: 1.) Using existing genome annotations (e.g., from repeatMasker[34]) to identify intervals of repetitive DNA and prohibiting design from these intervals (Stellaris Probe Designer, Chorus[20]); 2.)

Aligning efficacious probes outputted by upstream steps to the reference genome and discarding those aligning more than one or a few times (OligoArray[18], OligoMiner[22], ProbeDealer[24], Chorus[20], Chorus2[25]); 3.) Filtering efficacious probes outputted by upstream that contain the presence of *k*-mers with many occurrences in the reference genome (OligoMiner[22], iFISH[23], PROBER[19], Chorus2[25]). Collectively, these approaches are purpose-built to avoid designing probes that contain stretches of highly repetitive DNA, as such probes are an extremely problematic source of off-target background when the intended target is an interval of non-repetitive DNA[31,32]. However, if intervals of repetitive DNA are instead the intended design target, it is difficult or impossible to adopt existing workflows because of these filtering approaches, which are almost always hard-coded into the probe design tools. Specifically, approach 1 precludes the design of probes against any annotated repetitive region and thus cannot be present in any pipeline seeking to design probes against repetitive DNA intervals. Approach 2 is likewise problematic, as probes that target repetitive DNA by definition occur in the genome many times and thus would return many distinct alignment sites, but most probe design tools purposely filter any probe with more than 1 high-scoring alignment site. Finally, while approach 3 can be tweaked in some tools (*e.g.*, OligoMiner[22]) to allow probes containing high-occurrence *k*-mers to be designed, they do not contain any computational infrastructure to allow users to determine what proportion the occurrences of the relevant *k*-mers derive from the target interval; occurrences deriving from other genomic locations could lead to unacceptable levels of off-target background.

In order to help focus our tool development efforts, we reasoned that it would be helpful to start with a design test case. To this end, we turned our focus to the human alpha satellite repeat family, as it is present in the centromeric regions of all human chromosomes and most human chromosomes contain enough unique sequence variants to potentially allow chromosome-specific targeting[37]. Indeed, while chromosome-specific targeting of alpha satellite arrays with oligo probes has been reported for chromosome 17[55], alpha satellites are generally a challenging target for the design of interval-specific probes targeting just a single chromosome due to the high sequence similarities between the arrays of distinct chromosomes[56]; we have observed these challenges when using an oligo probe derived from chromosome 16 alpha satellite sequence[57], with which we observe staining on ~8–10 chromosomes[58,59]. For our test case, we focused on the alpha satellite arrays of chromosomes 2 (~2.4 Mb), 8 (~2.1 Mb), 21 (~349 kb), and Y (~317 kb). For each of these targets, we used a modified version of OligoMiner[22,26] to discover probes both from the entire array as well as the consensus monomers identified by Tandem Repeat Finder[60] (Methods). We next modified the PaintSHOP specificity profiling pipeline[26] to evaluate the predicted binding profiles of each designed probe (Methods). Our in silico predictions largely mirrored our empirical results with the chromosome 16 alpha satellite probe—of the 7897 probes we analyzed across the four targets, the vast majority had significant amounts of binding (>100 predicted binding sites) on the alpha satellite arrays of other chromosomes (Supplementary Fig. 1). Further analysis revealed that only a small proportion of the probes examined deriving from the full arrays (chr2: 57/4392, 1.3%; chr8: 268/2599, 10.3%; chr21: 6/600, 1.0%; chrY: 17/306, 5.6%) or consensus monomers (chr2: 1/14, 7.4%; chr8 1/32, 3.1%; chr21: 0/38, 0%; chrY: 0/3, 0%) only had predicted binding at their intended target (Supplementary Fig. 1).

Taken together, our analysis results suggest that for targets that share sequence similarity with related repeat family members such as human alpha satellites, in many cases it will be difficult to select a specific oligo probe by chance, even if such probes do exist for a given target. Moreover, while it is possible with an appropriate in silico analysis pipeline to pre-screen all possible probes to identify any specific probes for a given target, such an analysis requires considerable bioinformatic expertise to conduct and is computationally expensive, with the analysis of the four chosen alpha satellite arrays

collectively taking 512.6 h of CPU time (Supplementary Fig. 1) when run on a high-performance Linux cluster. With these results in mind, we set out to create a tool that would 1) use computationally efficient strategies to identify the probes that are the most likely to be specific and prioritize running those first to reduce the resource cost of the design process and 2) be wrapped in a user-friendly framework to allow researchers without deep expertise in bioinformatics to execute the design process.

## Oligo probe design with Tigerfish

Tigerfish is a computational pipeline composed of a collection of Python scripts embedded in an automated Snakemake workflow[61] that chains together novel code purpose built for Tigerfish and calls to existing bioinformatic tools that are commonly used to solve problems such as parallelized sequence alignment and *k*-mer counting. Tigerfish is designed to be executed in a command line environment. No direct knowledge of programming is required to run Tigerfish, and this bioinformatic workflow can be deployed on any modern Windows, Macintosh, or Linux system. Tigerfish is open-source, freely available via GitHub (https://github.com/beliveau-lab/TigerFISH), and depends on Bowtie2[28], NUPACK[62], Jellyfish[29], SamTools[63], Biopython[64], Scikit-learn[65], and chromoMap[66]. Tigerfish is also supported by extensive documentation (https://beliveau-lab-tigerfish.readthedocs-hosted.com). In order to run Tigerfish, users must include the full sequence of the genome assembly in which probe design is to be performed in FASTA format[67] and also provide an accompanying 'chrom.sizes' file that details the scaffolds present in the assembly and their lengths in base pairs. Users must also edit a small configuration file in which the locations of relevant files and scripts can be specified and parameter choices for the probe discovery can be specified.

Tigerfish can be run in one of three execution modes; these modes do not differ in the logic they use for designing and evaluating probes but allow different entry points into the process depending on what information the researcher already has in hand (Fig. 1). The first, termed "Repeat Discovery Mode", runs the full Tigerfish workflow end to end and is intended for cases in which researchers do not have a priori knowledge of where repetitive regions occur in their genome of interest. In Repeat Discover Mode, users list genomic scaffolds where de novo repeat discovery and probe design is to be performed in the configuration file. Repeat Discovery Mode uses a *k*-mer counting strategy to identify repetitive DNA regions de novo by identifying intervals that contain *k*-mers with high abundance in the genome (Methods). Users can tune the size of the search window and the magnitude of the *k*-mer count values needed for an interval to be flagged as repetitive, thereby controlling the nature of the repeat regions identified. The second, termed "Probe Design Mode", skips repeat discovery step and runs the Tigerfish pipeline starting from the probe design step (Fig. 1). Probe Design Mode is intended for instances where the genomic interval(s) a user wants to target for probe design are already known. In this case, the user must provide an additional BED-formatted file[68] that specifies the genomic coordinates for interval(s) to perform probe design against; these coordinates can refer to the full interval or, if available, the location of a consensus monomer. The third, termed "Probe Analysis Mode", runs the pipeline starting at the specificity filtering step that comes downstream of probe design (Fig. 1). Probe Analysis Mode is provided as a way to generate a new set of in silico binding predictions for probes contained in an existing Tigerfish output file; this functionality may be used to predict how the binding pattern of the input probes might change as the result of altering the salt concentration or melting temperature of the hybridization reaction. Tutorials providing a comprehensive walkthrough of these three modes, along with an example of implementing Tigerfish in the human T2T CHM13v2 + HG002 chrY genome assembly on a satellite repeat, can be found at https://beliveau-lab-tigerfish.readthedocs-hosted.com.

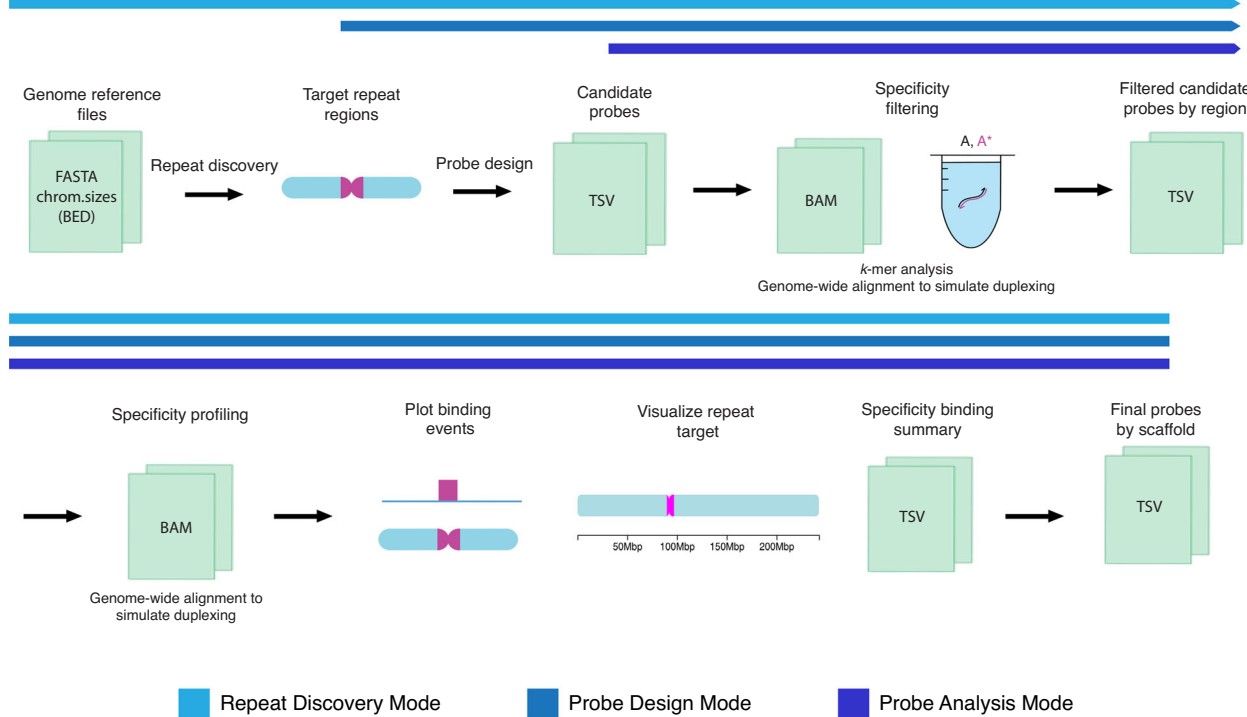

**Fig. 1 | The Tigerfish workflow.** Schematic overview of the inputs, major processing steps, and outputs of the Tigerfish probe design pipeline. Tigerfish may be run in three distinct modes, which progress through shared steps of the pipeline identically but enter at distinct steps.

When using Repeat Discovery Mode or Probe Design Mode, Tigerfish designs candidate oligo probes for each genomic interval passed forward (Repeat Discovery Mode) or specified in the user-provided BED (Probe Design Mode). Candidate probe discovery is performed using a modified version of the 'blockParse.py' script from OligoMiner[22] that screens the provided sequences for windows with desirable sequence and thermodynamic properties (Methods). To maximize the chance that the optimal probe or set of probes will be identified, Tigerfish mines the entire repeat region for candidate probes, which can result in redundant and even duplicate candidate probe sequences being returned. In order to minimize the amount of downstream computation needed, duplicates are removed and the candidate probes for each region are then rank-ordered to prioritize candidates that contain $k$-mers with elevated abundance specifically in the target interval from which they were designed (Methods), as such candidates are more likely to have many on-target binding sites while having minimal binding elsewhere in the genome.

In order to return a final probe set, Tigerfish begins with the top-ranked candidate probe for each target interval and performs deep in silico specificity profiling. The selected candidate probe is aligned to the genome with very sensitive settings (Methods) and up to 500,000 alignments are returned. The genomic sequence of each alignment site is then extracted and put into a virtual test tube to simulate how likely binding would be with the input candidate probe in FISH conditions using NUPACK[62]. Finally, Tigerfish processes the result of these simulations and calculates the number of predicted on- and off-target binding sites for each candidate probe (Methods). Users can specify a number of parameters to tune performance at this step, including the maximum number of allowed off-target binding sites per probe, the minimum number of required on-target binding sites per probe, and the maximum number of probes in the final set (Methods, Supplementary Note 1). If needed, Tigerfish will continue analyzing the predicted binding specificities of candidates from the rank-ordered list until either the user-supplied criteria are met or all possible candidate probes are considered. The final output of Tigerfish includes a text file

containing all final probes and their aggregate on- and off-target binding predictions, a summary table that lists all target intervals for which probes were designed and their aggregate on- and off-target binding predictions for the probes that map to each interval, and a set of auxiliary files that provide more detailed information about the predicted binding profiles of the probes. Users can also optionally populate chromoMap ideograms that depict the chromosomal locations of probe binding for the probe or set of probes designed against each target interval (Fig. 1). Example input and output files for full test runs of Tigerfish in Repeat Discovery Mode, Probe Design Mode, and Probe Analysis Mode can be found within Supplementary Software.

## Probe discovery at the scale of human genomes

In order to demonstrate the scalability of Tigerfish, we set out to perform genome-wide de novo repeat interval identification and probe design for all 24 chromosomes in the human T2T CHM13v2 + HG002 chrY assembly[36] using Repeat Discovery Mode. In order to showcase how users can tune parameters to optimize their design for different types of repeat regions, we performed our genome-scale runs with two sets of parameter groupings: 1) a 'conservative' set that prioritizes identifying large intervals of highly repetitive sequence such as those found at pericentromeres in order to prioritize extremely robust probes (> 500 target sites); 2) a 'permissive' set that aims to exhaustively discover intervals of repetitive DNA that can be probed with > 25 target sites (Supplementary Data 1). Analysis of our genome-wide probe design runs revealed that Tigerfish was able to design probe sets for 28 intervals using the 'conservative' parameter set (Supplementary Data 2) and 263 intervals using the 'permissive' parameter set (Supplementary Data 3). As neither parameter set puts an upper bound on the number of target sites or the size of the target interval, the 28 intervals discovered using 'conservative' parameter settings were also present discovered using 'permissive' parameter settings. We found that Tigerfish was able to generate at least one interval-specific probe or probe set for all 24 chromosomes, prominently covering the pericentromeric and subtelomeric regions of most

chromosomes. The Tigerfish probes mostly fell into regions not already covered by existing PaintSHOP probes[26] designed with non-repetitive intervals in mind (Fig. 2a) and predominantly mapped to annotated tandem and interspersed repeat families (Fig. 2b). We found that the repeat intervals identified spanned a broad range of sizes ranging from 411 bp to 34.3 Mb (median: 3.6 kb) for the group identified using the 'permissive' settings and from 37.6 kb to 34.2 Mb (median: 2.7 Mb) for the group identified using the 'conservative' settings (Fig. 2c). Collectively, these probes and probe sets cover 164.5 Mb of the human T2T CHM13v2 + HG002 chrY assembly after accounting for any differences between the size of the interval inputted for design and the effective size of the interval covered by the output probes (Supplementary Fig. 2).

Our in silico specificity profiling also revealed a broad distribution of the aggregate number of predicted on-target binding sites for the probes or probe sets covering the 263 intervals, ranging from 25 to 30,972 target sites in the 'permissive' group (median: 236.9 target sites) and 500–30,972 targets sites in the 'conservative group (median: 20,165.2 target sites) (Fig. 2d). When factoring in the size of the target intervals, we observed target site densities of 0.017–798.6 target sites per kb (median: 47.9 target sites per kb) for the 'permissive' group and 0.64–475.9 target sites per kb (median: 6.4 target sites per kb) for the 'conservative' group (Fig. 2e). Thus, the majority of the probe sets meet if not greatly exceed the threshold of ~200 target sites that in our experience leads to reliably robust DNA FISH; importantly, as this threshold is enforced by the efficiency of probe hybridization rather than absolute signal strength, we have observed that >200 probes is optimal even if signal amplification approaches such as Rolling Circle Amplification (RCA)[69], Hybridization Chain Reaction (HCR)[70], or Signal Amplification by Exchange Reaction (SABER)[71] are employed. Moreover, due to the reiterated nature of the target intervals, hundreds to many of thousands of target sites can be labeled by just one or a few probes, greatly reducing the cost of the FISH relative to experiments that target non-repetitive DNA with sets of hundreds to thousands of oligo probes. Thus, while Tigerfish is theoretically capable of designing probes against input intervals of any size, we recommend targeting intervals greater than 10 kb in length to maximize the chance of experimental success by providing > 200 binding sites for non-overlapping 40–50 nt probes; such regions are mostly to occur as tandem repeats and appear less frequently as interspersed repeats (Fig. 2b). On the level of the probes themselves, as we allowed a broad range of permissible lengths (25–50 nt), %G + C content (20–80%), and formamide-adjusting melting temperatures (42–52 °C), we observed a broad distribution of probe lengths (Fig. 2f) and formamide-adjusted melting temperatures (Fig. 2g), indicating that when Tigerfish is run with flexible probe parameter choices genome-wide it is able to design probes suitable for the variable nucleotide compositions found at it repetitive targets.

## Validating Tigerfish probes in situ

In order to evaluate how effectively the in silico design approach of Tigerfish translates to performance in situ, we designed and conducted a series of FISH experiments. Specifically, we set out to investigate whether Tigerfish was able to generate a panel of FISH probes targeting repetitive DNA intervals specific to each of the 24 human chromosomes, as such a panel would have utility in diagnostic and chromosomal enumeration assays. In order to showcase the versatility of the different Tigerfish run modes, our panel consisted of a mix of probes designed against regions identified using "Repeat Discovery Mode" and regions selected manually based on their RepeatMasker[34] annotations using "Probe Design Mode" (Supplementary Data 4 and Supplementary Data 5). The panel spanned a range of target sizes (10 kb–4.5 Mb, mean = 1.3 Mb) and predicted on-target binding activities (477.5–7228, mean = 2418.1) and includes a mix of tandem and interspersed repeat targets (Table 1). Of the tandem repeats, chromosome-specific alpha satellite probes were designed for

chromosomes 2–12, 17–18, 20, and X (Table 1). In order to verify that our Tigerfish probes were binding to their intended genomic targets, we implemented an experimental scheme in which the Tigerfish probe set targeting a given interval was co-hybridized with a set of 1000 probes designed by PaintSHOP[26] that targeted a 200 kb non-repetitive interval on the target chromosome, with the Tigerfish and PaintSHOP probe sets being labeled with spectrally distinct fluorophores (Fig. 3a, Supplementary Data 5). We used this experimental design to perform a series of 24 two-color FISH experiments on 46, XY human primary metaphase chromosome spreads (Fig. 3b). Using this approach, we confirmed that our metaphase FISH produced the predicted staining patterns for all 24 combinations of Tigerfish and PaintSHOP probe sets (Fig. 3c, Supplementary Figs. 3–6). In order to augment our metaphase data with a sample type that provides a means to visually inspect that we were achieving the correct staining pattern with a larger sample size, we also performed a series of 24 interphase FISH experiments on 46, XY primary human lymphoblasts using the same Tigerfish and PaintSHOP probe set combinations as a means to visually enumerate chromosomal copy number (Fig. 4a). Specifically, we imaged > 40 cells for each experiment and quantified the number of observed Tigerfish and PaintSHOP foci in the 3D volume of the nucleus (Fig. 4b, Supplementary Fig. 7–10). Our analysis of the resulting data revealed a strong agreement between the two types of probe set (78.4% concordance, $n = 1061$), with both approaches predominantly displaying 2 foci per nucleus (PaintSHOP: 781/1061, 73.6%; Tigerfish: 922/1061, 86.9%) and identifying a range of foci (1–4) per nucleus consistent with our previous studies using oligo-based probes for enumeration[10,22,72] (Fig. 4c). We also noted that we consistently identified more Tigerfish foci than PaintSHOP foci. Although we cannot formally rule out differences in the underlying copy number of the target loci or frequency of sister chromatid separation as the two probe set types target distinct chromosomal regions, we believe the most likely explanation for this observation is that the PaintSHOP probes required three rounds of hybridization (primary oligo library, secondary PER-extended "bridge" oligos, tertiary fluorescently labeled "imager" oligos), while the Tigerfish probes only required two (primary PER-extended probe oligos, secondary imager oligos). We and others have observed reduced labeling efficiency due to the additional hybridization round required to use bridge oligos[12–14,73], and further support for this observation comes from the higher frequency of nuclei with no detectable PaintSHOP foci relative to the number with no detectable Tigerfish foci (Tigerfish: 13/1061, 1.2%; PaintSHOP: 48/1061, 4.5%; two-sided Fisher's exact $p$-value = $5.7 \times 10^{-6}$). Taken together, our metaphase and interphase FISH experiments demonstrate the specificity and utility of Tigerfish for visualizing the positioning and abundance of highly repetitive DNA intervals in situ.

## Computational requirements to run Tigerfish

In order to evaluate the computational resources required to run Tigerfish at the scale of mammalian genomes, we collected a series of benchmarking data during our probe design runs on the full human T2T CHM13v2 + HG002 chrY assembly using the 'permissive' and 'conservative' parameter settings. Our analyses focused on four key usage metrics: 1) the "wall clock" run time, which reflects the overall duration of the run from start to finish; 2) the amount of active CPU processing time needed to complete the run; 3) the maximum amount of virtual memory used, which represents the sum total of physical (RAM) and swap (hard disk) memory allocations; 4) the maximum amount of physical memory used, which reflects the RAM component of the virtual memory pool. As Tigerfish uses Snakemake[61] for parallelization, we were able to record data about these four metrics on a per-interval basis for all 263 intervals identified collectively by the 'permissive' and 'conservative' parameter settings. In line with the broad range of observed target interval sizes and target site numbers of the 263 intervals (Fig. 2f–h), we also found a wide distribution of

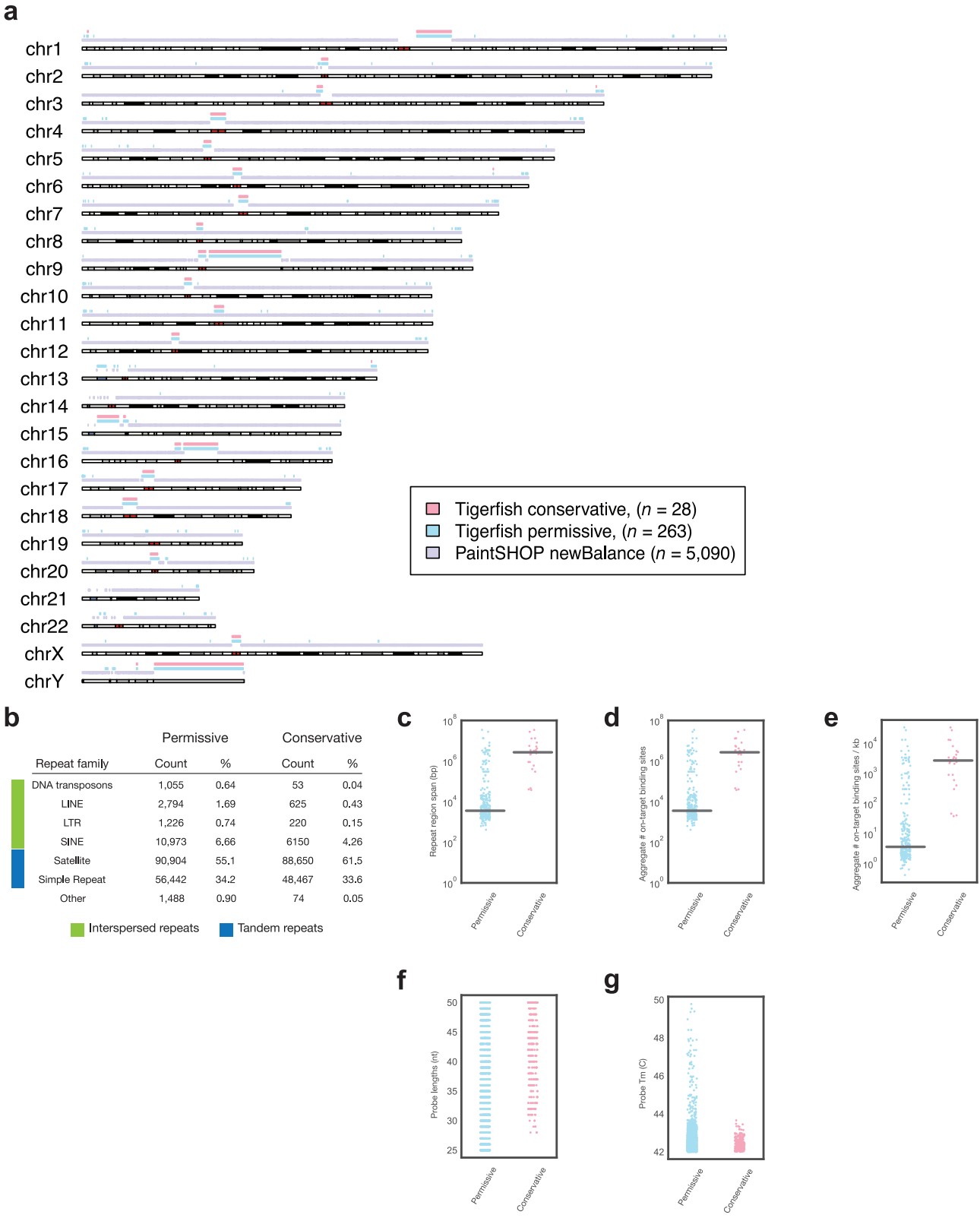

**Fig. 2 | Genome-scale probe design with Tigerfish. a** Schematic visualization of intervals for which Tigerfish probe sets were identified using conservative (pink) or permissive (teal) parameters and intervals covered by existing PaintSHOP probes designed using parameters suitable for non-repetitive targets (lilac). **b** The distribution of RepeatMasker annotations for intervals identified and processed by Tigerfish using conservative and permissive settings. **c** Length distributions of the regions identified and targeted by Tigerfish using conservative and permissive parameters. **d** The aggregate number on-target binding predictions for probe sets designed by Tigerfish using conservative and permissive parameters. **e** The aggregate number on-target binding predictions per kilobase for probe sets designed by Tigerfish using conservative and permissive parameters. **f** The distribution of the lengths of the probes discovered using conservative and permissive parameters. **g** The distribution of the melting temperatures of the probes discovered using conservative and permissive parameters. The sample sizes for (**c**–**e**) are $n = 263$ intervals for the Permissive dataset and $n = 28$ intervals for the Conservative dataset. The sample sizes for (**f, g**) are $n = 1954$ probes for the Permissive dataset and $n = 286$ probes for the Conservative dataset.

**Table 1 | Description of the 24-target Tigerfish probe set panel**

| Imaging Coordinates | On-target | Off-target | Imaging Repeat Length (Mb) | Annotation |
|---|---|---|---|---|
| chr1:134680000–134800000 | 7228.4 | 191.2 | 0.12 | Satellite (hsat2) |
| chr2:92330000–94670000 | 3174.8 | 243.0 | 2.34 | Satellite (alpha) |
| chr3:91730000–92590000 | 505.4 | 161.4 | 0.86 | Satellite (alpha) |
| chr4:52140000–53070000 | 1074.6 | 51.0 | 0.93 | Satellite (alpha) |
| chr5:47650000–48150000 | 1675.0 | 352.1 | 0.5 | Satellite (alpha) |
| chr6:58540000–61060000 | 2356.1 | 73.6 | 2.52 | Satellite (alpha) |
| chr7:60410000–63720000 | 4977.8 | 251.6 | 3.31 | Satellite (alpha) |
| chr8:44250000–46320000 | 1964.4 | 933.4 | 2.07 | Satellite (alpha) |
| chr9:44960000–47230000 | 2049.0 | 261.9 | 2.27 | Satellite (alpha) |
| chr10:39640000–40710000 | 1637.4 | 25.3 | 1.07 | Satellite (alpha) |
| chr11:51040000–54420000 | 3908.9 | 110.8 | 3.38 | Satellite (alpha) |
| chr12:34780000–37060000 | 2022.9 | 108.8 | 2.28 | Satellite (alpha) |
| chr13:111520000–111570000 | 927.9 | 644.1 | 0.05 | Novel |
| chr14:99470000–99490000 | 477.5 | 1188.7 | 0.02 | Novel |
| chr15:8550000–8680000 | 4162.8 | 1608.9 | 0.13 | Satellite (hsat3) |
| chr16:48950000–48980000 | 3812.7 | 807.1 | 0.03 | Satellite (hsat2) |
| chr17:23890000–27420000 | 3912.9 | 512.0 | 3.53 | Satellite (alpha) |
| chr18:15970000–20430000 | 4576.7 | 3319.3 | 4.46 | Satellite (alpha) |
| chr19:21000000–21060000 | 1408.6 | 261.4 | 0.06 | Satellite (centromeric) |
| chr20:27580000–27630000 | 950.1 | 174.9 | 0.05 | Satellite (alpha) |
| chr21:44760000–44780000 | 761.1 | 440.1 | 0.02 | Complex (LINE, SINE, DNA transposons) |
| chr22:18540000–18550000 | 1347.7 | 295.7 | 0.01 | Complex (Simple Repeat, SINE) |
| chrX:58910000–59080000 | 1518.8 | 146.5 | 0.17 | Satellite (alpha) |
| chrY:20960000–21230000 | 1603.9 | 175.5 | 0.27 | Satellite (centromeric) |

resource usage values. Our analyses revealed that probe design against the majority of target intervals finished quickly, with a median run time of 6.9 min (range: 1.8 min–50.2 h) and a median CPU time of 5.2 min (range: 0.6 min–4.9 h) (Fig. 5a, b). Moreover, Tigerfish generally required only modest amounts of memory for software designed to be run on a computing cluster, with a median max virtual memory allocation of 24.8 GB (range 12.3–44.8 GB) and a median max physical memory allocation of 20.3 GB (range 8.2–40.8 GB) (Fig. 5c, d). Given the observed spread in the resource usage values, we hypothesized that the resource requirements might vary as a function of the size of the target interval. Indeed, stratifying the benchmarking data into three groups based on span of the target interval revealed that the group of intervals less than 100 kb in span had a median run time of 5.8 min (range: 1.8 min–43.8 min, $n = 211$) and the group of intervals between 100 kb and 1 Mb had run a median run time of 21.6 min (range: 6 min–59.4 min, $n = 20$), with the group of intervals >1 Mb in span having a considerably longer median run time of 4.6 h (range: 16.2 min–50.1 h, $n = 32$) (Supplementary Fig. 11). We did not observe a similar trend with virtual memory or physical memory usage, as all three length groups had nearly identical memory requirements (Supplementary Fig. 11). As large genomic targets often will contain many possible candidate probes, it may take Tigerfish some time to arrive on one or more suitable probes that satisfy the user-specified requirements for the aggregate on-target binding while only including probes with low predicted off-target binding activity at such targets. While 1–2 day compute runs are often standard for genome-scale probe design tasks in large and complex genomes[22,26], users may still attempt to reduce the processing time for problematic intervals by adjusting the parameters used for the rank-ordering of candidate probes or relaxing the probe design requirements (Supplementary Note). Collectively, our benchmarking results indicate that Tigerfish can readily be deployed on computing clusters or powerful individual computers to identify repetitive intervals and design probes specific to these intervals at the scale of genomes.

## Discussion

Tigerfish is a freely available computational platform that facilitates the design of oligo-based FISH probes against intervals of repetitive DNA at the scale of genomes. The Tigerfish pipeline establishes a paradigm for the deep specificity analysis of probes targeting repetitive sequences, which in turn enables users to establish criteria by which to select and empirically evaluate the effectiveness of oligos targeting such regions. Once designed, Tigerfish probes can readily be augmented with any of the powerful toolkits available for oligo-based FISH, including signal amplification approaches such as SABER[71], HCR[70], and RCA[69] and multiplexing approaches such as DNA MERFISH[74] and DNA seqFISH[14]. Moreover, Tigerfish offers users a great number of tunable parameters, providing flexibility to tailor the probe design process for different types of repetitive intervals and different genome compositions and complexities. We have demonstrated the efficacy of Tigerfish by performing genome-scale probe discovery in a fully assembled human genome and provided extensive experimental validation on both spread metaphase chromosomes and in interphase nuclei for the specificity of Tigerfish probes. Tigerfish is supported by extensive documentation and tutorials and can perform complex probe discovery tasks against the most challenging intervals of genomic DNA using only modest computational resources.

Like all tools, Tigerfish has limitations and avenues for future development. For instance, the command-line nature of Tigerfish and its optimization for cluster-based computing may prove to be a barrier to entry for some users. Future work to implement a graphical user interface and to provide a cloud-based platform for running Tigerfish may help to make it accessible to a broader set of researchers. Tigerfish's ability to design probes against specific intervals of highly repetitive DNA is also highly dependent on the quality of input genome assembly, and to date there are only a few fully polished telomere to telomere assemblies optimal for Tigerfish deployment. Nevertheless, we anticipate Tigerfish will play a key role in the experimental validation and biological investigation of repetitive DNA intervals as more

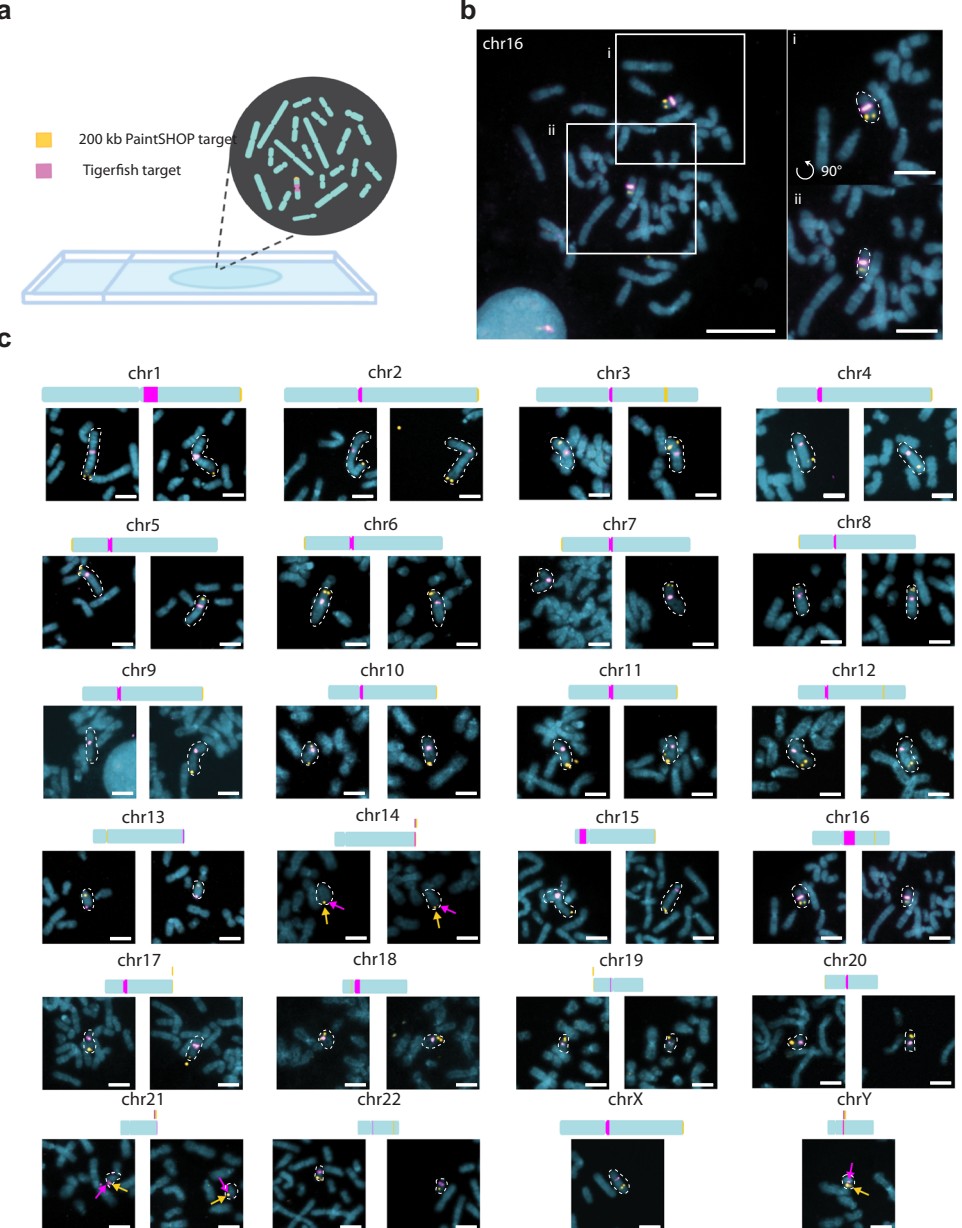

**Fig. 3 | In situ validation of Tigerfish probes. a** Schematic overview of the experimental design used to validate Tigerfish probe sets on metaphase chromosome spreads also labeled with probe sets targeting non-repetitive DNA designed by PaintSHOP. **b** Representative full field of view (left) and zoomed insets (right) showing Tigerfish (magenta) and PaintSHOP (yellow) probe sets targeting chr16. **c** Zoomed crops depicting Tigerfish (magenta) and PaintSHOP probes targeting the indicated chromosomes. For the autosomes, each image pair was obtained from the same metaphase spread. The X and Y chromosome images were obtained from separate 46, XY spreads and thus only have one chromosome each. Please see Supplementary Figs. 3–6 for the full spread images. Images are maximum intensity projections in Z. Scale bars, 5 μm (zoomed crops) or 20 μm (fields of view). Each staining pattern was visualized in three independent samples and yielded similar results.

fully assembled human, vertebrate, plant, and other model organism genomes continue to be introduced.

## Methods
### Data collection
Nikon Elements AR 5.20.00 was used to acquire microscopy images on the Nikon Ti-2 system.

### Data analysis
Tigerfish probe design was performed with open source code hosted here: https://github.com/beliveau-lab/TigerFISH. Tigerfish is written in Python 3.7.8 with dependencies that include Biopython 1.77, Bowtie 2.3.5.1, NUPACK 4.0, BEDtools 2.29.2, Numpy 1.18.5, Pandas 1.0.5, pip 20.1.1, pybedtools 0.8.1, sam2pairwise 1.0.0, samtools 1.9, scikit-learn 0.23.1, scipy 1.5.0, zip 3.0, matplotlib 3.3.4, seaborn 0.11.1, pytest 6.2, and Jellyfish 2.2.10. All Tigerfish probe collections were generated using a pipeline implemented with Snakemake 7.19.

### Genome sequences used for probe set design
The CHM13 genome assembly versions 1.0, 1.1, and 2 ( + HG002 chrY) were downloaded without repeat masking from the T2T consortium at https://github.com/marbl/CHM13.

### Design and analysis of human alpha satellite probes
Sequence annotations for the locations of the alpha satellite arrays located on human chromosomes 2, 8, 21, and Y was obtained from the

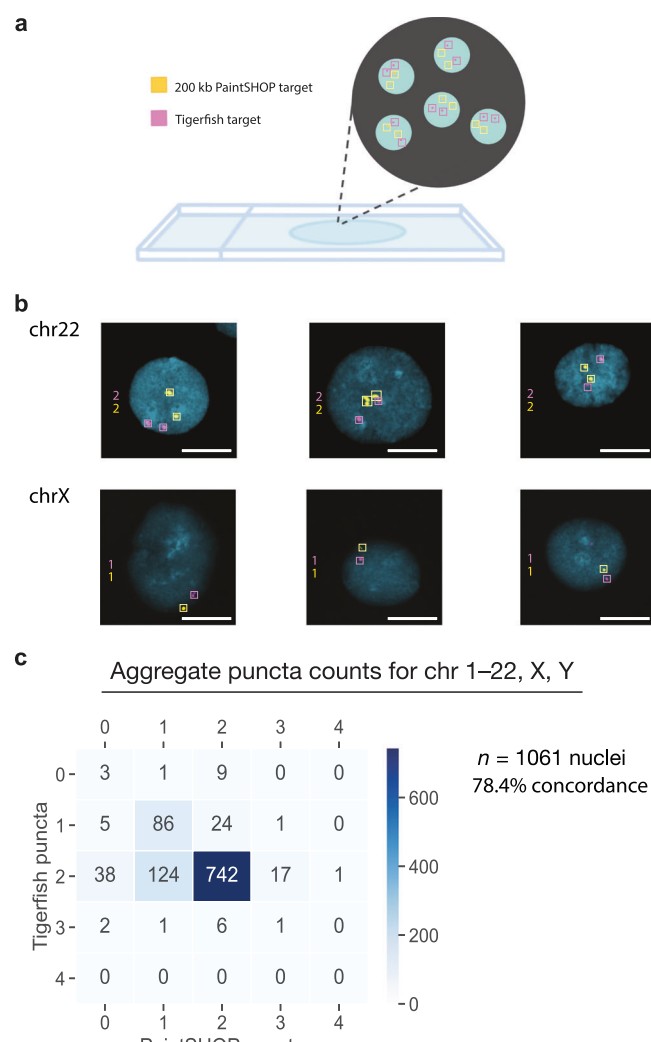

**Fig. 4 | Chromosome enumeration in interphase nuclei. a** Schematic overview of the experimental design used to perform chromosome enumeration using Tigerfish probe sets in 46, XY interphase nuclei also labeled with probe sets targeting non-repetitive DNA designed by PaintSHOP. **b** Representative images of nuclei labeled with Tigerfish probe sets (magenta) and PaintSHOP probe sets (yellow) targeting intervals on chr22 (top row) or chrX (bottom row). **c** Heatmap displaying the observed distribution of Tigerfish and PaintSHOP puncta per nucleus. The data presented are aggregated over all the 24 chromosomes targeted. Nuclei with >2 puncta most likely contain replicated sister chromatids that have separated. Images are maximum intensity projections in Z. Scale bars, 10 μm. Each staining pattern was visualized in three independent samples and yielded similar results.

T2T consortium at https://github.com/marbl/CHM13 in CHM13v2.0 coordinates. Consensus monomers were identified using Tandem Repeat Finder 4.09 using the following command string: "trf *input.fa* 2 5 7 80 10 50 2000". Oligo probe sequences were discovered using a modified version of the OligoMiner "blockParse.py" script[22,26] with the parameters "-l 25, -L 50, -t 42 -T 52". Predicted binding profiles were generated by aligning probe sequences to the T2T CHM13v2 + HG002 chrY using Bowtie2[28] with the parameters "−local -N 1 -R 3 -D 20 -i C,4 −score-min G,1,4, -L 15, k 500000". The genomic sequences of the alignment sites returned by Bowtie2 was extracted by sam2pairwise and evaluated for tested for thermodynamic stability with the input probe sequence using NUPACK[62,75] with the parameters "material='dna', celsius=69.5, sodium=0.39, magnesium=0.0, ensemble = 'stacking'". Alignments and their associated thermodynamic binding predictions were mapped back to alpha satellite arrays using BEDtools[76]. Resource tracking was performed using Snakemake[61].

## Pipeline construction and implementation

Tigerfish is written in Python 3.7.8 with dependencies that include Biopython 1.77[64], Bowtie 2.3.5.1[28], NUPACK 4.0[62,75], BEDtools 2.29.2[76], Numpy 1.18.5[77], Pandas 1.0.5[78], pip 20.1.1, pybedtools 0.8.1[76,79], sam2-pairwise 1.0.0[80], samtools 1.9[63], scikit-learn 0.23.1[65], scipy 1.5.0[81], zip 3.0, matplotlib 3.3.4[82], seaborn 0.11.1[83], pytest 6.2[84], and Jellyfish 2.2.10[29]. All Tigerfish probe collections were generated using a pipeline implemented with Snakemake 7.19[61]. Dependencies that implement Python libraries can be found via the tigerfish.yml, snakemake_env.yml, and chromomap_env.yml files that are used to execute Tigerfish as a Snakemake[61] pipeline. These scripts and their dependencies are documented on Tigerfish's GitHub repository. These environments are also available in the Supplementary Software. Scripts were executed locally in an OS X Anaconda Python 3.7[85] environment or in a CentOS Linux environment on the Department of Genome Science 'Grid' Cluster at the University of Washington.

## Whole genome probe discovery

Genome assemblies in FASTA format without repeat masking were used when building Jellyfish[29] files and Bowtie2[28] indices, and were used as input files for probe discovery. Jellyfish hash size was set to approximate the size of the genome assembly so that files were generated using the command, "jellyfish count -s 3300 M -m 18".

## Identification of *k*-mer enriched sequences

Tigerfish identifies repeat regions in Repeat Identification mode by using a sliding window of a specified size (*window*, W) flagging all counts exceeding a user-specified value (*threshold, T*). The sum of the counts within the sliding window are divided by the length of the window so that as long as the user-specified composition score (*composition, C*) is exceeded, Tigerfish will identify windows of the genome where k-mer counts which map to abundantly repetitive sequences. Here, users may also specify at what base position they wish to start searching for repeats, which is described as a *file_start* parameter. Alternatively, if the user provides coordinates of target regions (i.e., defined_coords=True and repeat_discovery=False), then the user must also provide the name of the scaffold. In this case, Tigerfish skips the 'repeat_ID.py' script entirely to proceed with oligo probe design. For whole genome mining in the T2T CHM13v2 + HG002 chrY assembly, the sliding window was implemented with parameters described in Supplementary Data 1.

## Designing oligo probes

Tigerfish implements logic as described in the OligoMiner[22] framework for probe design using the bed file generated during Repeat Identification mode or from a user-provided BED file. Here, a FASTA file containing all regions of interest is used to design valid probe sequences using parameters values for probe length, percent G + C content (GC%) and adjusted melting temperature $T_m$ calculated using nearest neighbor thermodynamics[22]. The modified blockParse script described in OligoMiner was used to mine probe candidates ranging in length from (*min_length, max_length*) 25–50 nt and $T_m$ (*min_temp, max_temp*) between 42 and 52 °C.

## Predicting probe specificity

The *k*-mer binding proportion (*enrich_score*, $K_b$) was determined by obtaining the proportion of two computed values, *copy_num* and *total_genome_binding*. The aggregate count of all *k*-mers for any given probe sequence within its respective repeat target is described as *copy_num*, or $R_m$. The aggregate count of all *k*-mers for any given probe within the entire queried genome is described as *total_genome_binding*, or ($H_m$). Thus, the *k*-mer binding proportion was computed as $R_m$/$H_m$. Probes with shared *k*-mer composition similarity above the *mer_cutoff* proportion are omitted from downstream filtering. Probes are ranked in descending order within each repeat region by Normalized

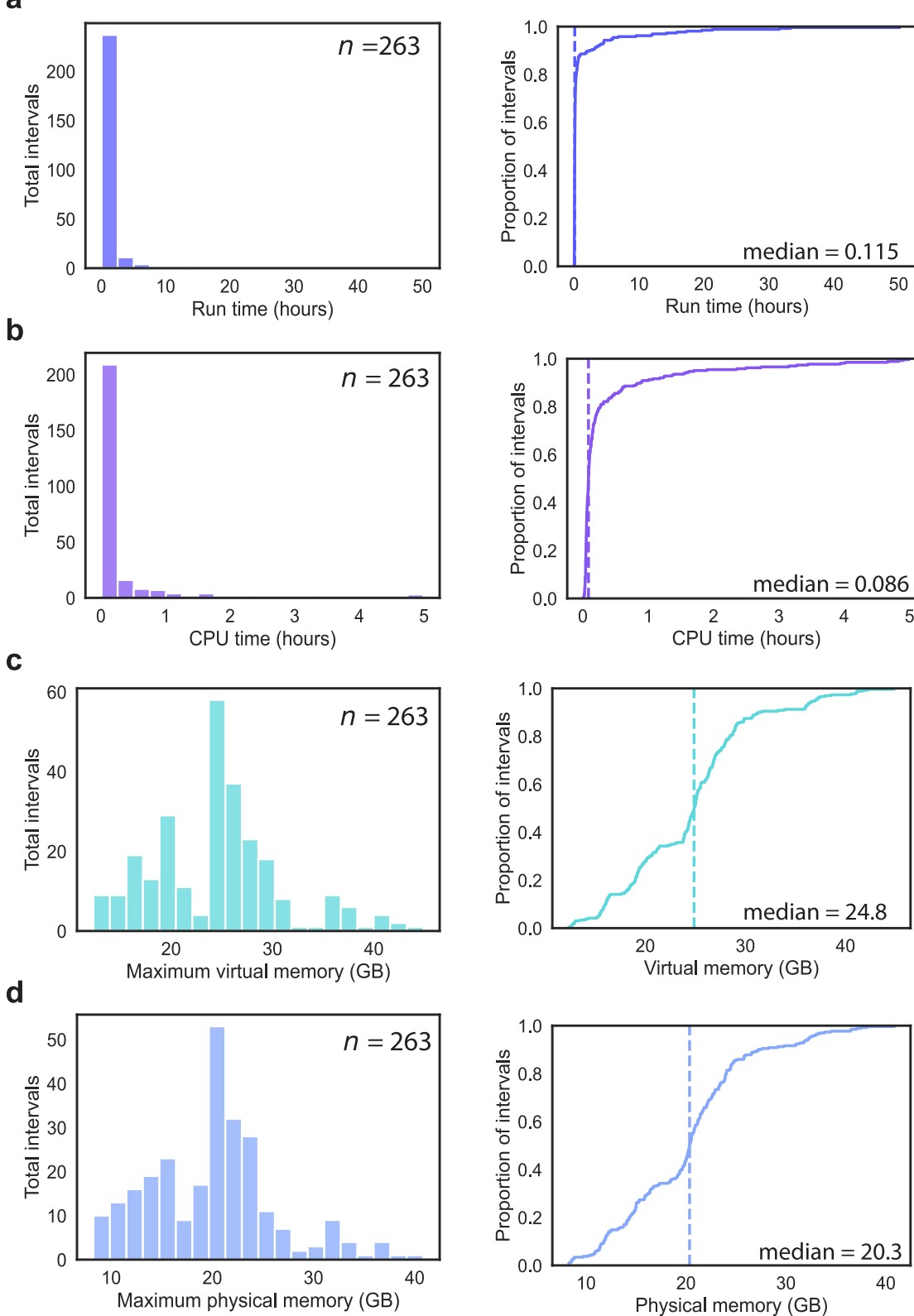

**Fig. 5 | Resource requirements for genome-scale Tigerfish probe design.**
**a** Distribution (left) and empirical cumulative distribution (right) of the wall-clock runtime recorded for running the 263 conservative and permissive intervals.
**b** Distribution (left) and empirical cumulative distribution (right) of the CPU runtime recorded for running the 263 conservative and permissive intervals.
**c** Distribution (left) and empirical cumulative distribution (right) of the maximum recorded virtual memory allocation for running the 263 conservative and permissive intervals. **d** Distribution (left) and empirical cumulative distribution (right) of the maximum recorded physical memory allocation for running the 263 conservative and permissive intervals. Vertical dashed lines in the cumulative distribution plots correspond to the median values.

Rank $(N_r) = (R_m/(\max(R_m)*c1)) + (K_b/(\max(K_b)*c2))$, where c1 (*c1_val*) and c2 (*c2_val*) are user-specified constants.. The *mer_cutoff* proportion is determined by storing *k*-mers of ranked probes and profiling all consecutive candidate probes to see if the proportion of their *k*-mer composition exceeds that of the *mer_cutoff*. Users may modify *enrich_score*, *copy_num*, *c1_val*, *c2_val*, and *mer_cutoff* within the config.yml file. The parameters chosen for the conservative and permissive datasets are reported in Supplementary Data 1.

### Computing in silico binding predictions

Bowtie2 was run on each probe sequence against the human genome using the following parameters (−local -N 1 -R 3 -D 20 -i C,4 −score-min G,1,4, -L 15, k 500000). The parameters -L (*seed_length*) and -k (*bt2_alignment*) may be modified by users within the Tigerfish config.yml. Probe alignments are returned as a BAM file for each probe sequence, which is then processed from the resulting SAM file using SAMtools[63]. Using this SAM file, sam2pairwise[80] is used to return derived alignment sequences. With these provided pairs of probe sequence and derived alignment sequence, NUPACK 4.0[62] computes the predicted thermodynamic likelihood that each alignment pair will form duplexes under FISH conditions[22]. The NUPACK model summarizing these conditions is described as (material='dna', celsius=69.5, sodium=0.39, magnesium=0.0, ensemble = 'stacking'). Candidate probes are only added to the final probe set if they do not share predicted probe binding greater than the value *max_pdups_binding*, which is defined as the maximum predicted binding probability between a given candidate and any probes already selected for the target interval.

The on-target alignment score $(On_T)$ is determined by taking the sum of all predicted duplexing scores for derived alignments that are found within the repeat target. Off-target alignment scores are computed by taking the aggregate sum of all predicted duplexing scores from derived alignments that are found outside the repeat target $(Off_T)$. The predicted in silico on-target binding proportion (*binding_prop*) for each oligo is then computed as $On_T/(On_T+Off_T)$. Genome bins (*genome_windows*) are generated using BEDtools makewindows, and BEDtools intersect is applied to all reported sam2pairwise genome alignments to identify potential off-target binding signals. All predicted duplexing scores are aggregated over windows, which are binarized to map binding signals to the repeat target and all other genomic regions where binding events are predicted. Probes with an aggregate $Off_T$ over any given non-target genome bin that exceeds the parameter *off_bin_thresh* are culled from the candidate probe set. Users may modify the parameters seed_length, bt2_alignment, genome_windows, binding_prop, and off_bin_thresh. There are additional parameters that may be used to control permissiveness of filtering in the alignment_filter.py script. Users may control the desired aggregate on-target sum for any set of probes designed against a repeat region (*target_sum*), the minimum on-target value for any desired candidate probe (*min_on_target*), and maximum desired candidate probes to be returned in any target repeat region (*max_probe_return*). Parameters chosen for conservative and permissive datasets may be viewed in Supplementary Data 5.

### Visualizing candidate probe in silico binding

Bowtie2 alignments are derived for individual or pools of probes against a repeat region where predicted thermodynamic binding is computed over a given size of genomic bins generated by BEDtools (*thresh_windows*). These predicted thermodynamic binding events are summarized by scaffolds and are used to determine the size of the imaging target window for bins containing binding events that are greater than the parameter within the repeat region target (*align_thresh*). The sum of predicted duplexing values are aggregated over computed genomic bins and normalized using the MinMaxScalar function of scikit-learn[86], where the range of values is mapped from 0 to 255 to summarize predicted binding over genomic bins.

chromoMap[87] in R is used to generate summary ideograms of probe target signals as an optional step in Probe Analysis Mode.

### Read the Docs

A Read the Docs web page (https://beliveau-lab-tigerfish.readthedocs-hosted.com) was created to provide detailed documentation of our tool. The intention of hosting our work on Read the Docs was to provide sufficient background and resources for individuals from all computational backgrounds to be able to leverage Tigerfish for their own work. Here we provide installation information, simple tutorials for testing the Tigerfish install, a glossary of all parameters that may be modified by users, summaries of our default parameters, and frequently asked questions.

### Computational benchmarking

Speed calculations were computed using the Snakemake benchmark feature. Each scaffold in the T2T CHM13v2 + HG002 chrY assembly was run as its own individual cluster job in parallel for the repeat discovery steps, and the resulting intervals identified for probe design were also processed in parallel. Benchmarking was performed on a Dell PowerEdge R840 server node equipped with 4 Intel Xeon Gold 6252 2.1 GHz 24-core CPUs (192 total job threads) and 1.5 TB of DDR4 PC4-23400 2933 Mhz ECC RAM running CentOS 7.9 Linux.

### PER concatemerization

100 µl Primer Exchange Reactions were prepared for both Tigerfish probes and PaintSHOP bridge sequences with a final concentration of 1x PBS, 10 mM MgSO4, 400–1000 U/ml Bst DNA Polymerase (large fragment), 120,000 units/ml (NEB M0275M), 100 nM Clean G hairpin, 50 nM–1 µM hairpin and water to 90 µl. After incubation for 15 min at 37 °C, 10 µM oligo probe(s) were added and the reaction was incubated for another 2 h with another 20 min at 80 °C to heat-inactivate the polymerase. PER extension solutions were directly diluted into FISH solutions. Lengths of the concatemers were evaluated by diluting 6.7 µl of the in vitro reaction with 3.3 µl 6X TriTrack. Samples were then run on a 10% TBE-Urea denaturation gel (ThermoFisher EC68755BOX) for 10 min alongside 1 kb Plus DNA Ladder to estimate length and imaged with SYBR Gold channel and then imaged after a 15 min incubation.

### DNA-SABER-FISH on spread metaphase chromosomes and interphase nuclei

PaintSHOP bridge oligos and Tigerfish primary probes were extended using the PER as previously described[71] and described above in "PER concatemerization". Dry microscope slides containing human 46,XY metaphase spreads and intact lymphoblastoid nuclei (Applied Genetics Laboratories, catalog #s HFM and HMM) and intact interphase nuclei were immersed in 2× SSCT + 70% (vol/vol) formamide at 70 °C and incubated for 90 s in Coplin jars. Slides were then transferred and incubated in ice-cold 70% (vol/vol) ethanol, ice-cold 90% (vol/vol) ethanol, and ice-cold 100% (vol/vol) for 5 min each. Slides were then air dried after incubation in 100% ethanol. A hybridization solution consisting of 2X SSCT, 50% formamide, 10% (wt/vol) dextran sulfate, 40 ng/µL RNase A (EN0531; Thermo Fisher), and resuspended PER-extended PaintSHOP bridge oligos (20 pmol total), amplified ssDNA primary probes (25 pmol total), and PaintSHOP bridge library (60 pmol total) which were dried at 60 °C for 30 min using a SpeedVac concentrator. The solution was sealed using a 22 × 22 mm #1.5 coverslip using rubber cement. Samples hybridized overnight at 45 °C in a humidified chamber. Samples were then washed for 15 min in 2X SSCT at 60 °C and then twice for 5 min with room temperature 2X SSCT. Samples were then incubated in a secondary hybridization containing 5X PBST, 10% dextran sulfate, 10 µM fluorescent oligos for 1 h at 37 °C. Slides were then washed three times with 1X PBST at 37 °C. After air drying slides, samples were mounted with SlowFade Gold + DAPI and sealed beneath a 22 × 30 mm #1.5 coverslip using nail polish.

## Microscopy

Microscopy was performed using a Yokogawa CSU-W1 SoRa spinning disc confocal unit attached to a Nikon Eclipse Ti-2 chassis. Excitation light was emitted at 30% of maximal intensity from 405 nm, 488 nm, 561 nm, or 640 nm lasers housed inside of a commercial Nikon LUNF 405/488/561/640NM launch. Laser excitation was delivered via a single-mode optical fiber into the CSU-W1 SoRa unit. Excitation light was then directed through a microlens array disc and a 'SoRa' disc containing 50 μm pinholes and directed to the rear aperture of a 100x N.A. 1.49 Apo TIRF oil immersion objective lens by a prism in the base of the Ti2. Emission light was collected by the same objective and passed via a prism in the base of the Ti2 back into the SoRa unit, where it was relayed by a 1x lens through the pinhole disc and directed into the emission path by a quad-band dichroic mirror (Semrock Di01-T405/488/568/647-13x15x0.5). Emission light was then spectrally filtered by one of four single bandpass filters (DAPI: Chroma ET455/50 M; ATTO 488: Chroma ET525/36 M; ATTO 565: 27 Chroma ET605/50 M; Alexa Fluor 647: Chroma ET705/72 M) and focused by a 1x relay lens onto an Andor Sona 4.2B-11 camera with a physical pixel size of 11 μm, resulting in an effective pixel size of 110 nm. The Sona was operated in 30 16-bit mode with rolling shutter readout and exposure times of 300 ms. Images were processed in ImageJ and Fiji[88,89] and Adobe Photoshop.

## Reporting summary

Further information on research design is available in the Nature Portfolio Reporting Summary linked to this article.

## Data availability

The primary bioinformatic data can be accessed via the Source Data file that accompanies this manuscript. Primary microscopy data will be made available upon request. The CHM13 genome assembly versions 1.0, 1.1, and 2 (+HG002 chrY) can be downloaded without repeat masking from the T2T consortium at https://github.com/marbl/CHM13. Source data are provided with this paper.

## Code availability

The Tigerfish source code is available under a MIT license at https://github.com/beliveau-lab/TigerFISH.

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

## Acknowledgements

We thank Dr. Evan Eichler for providing CHM13-hTERT cells and Dr. Tamara Potapova and Dr. Jennifer Gerton for their valuable advice for preparing fixed metaphase slides. Additionally, we would like to thank Dr. Ching-Ho Chang and Dr. Amanda Laurracuente for their feedback on the *D. melanogaster* genome. We thank David Nwizugbo, Caleb Kono, Caleb J. Bower, and Chris Hsu for feedback on the development of Tigerfish and members of the Beliveau and Noble labs for helpful discussions. This work was supported by the National Institutes of Health (under grants 1R35GM137916 to B.J.B., UM1HG011531 to W.S.N., and 1R01HG011274 to K.H.M.) and the Brotman Baty Institute for Precision Medicine (under a Catalytic Collaboration award to B.J.B.). R.A. was supported by a National Science Foundation Graduate Research Fellowship Program Award and a Howard Hughes Medical Institute Gilliam Fellowship for Advanced Study. C.K.C. was supported by NIH training grant 5T32HG000035.

## Author contributions

R.A., W.S.N., and B.J.B. conceived the study. R.A., C.K.C., and Q.L. wrote and optimized software code. R.A. and C.K.C. performed experiments. R.A., W.S.N., and B.J.B. wrote the manuscript. K.H.M., W.S.N., and B.J.B. supervised the work.

## Competing interests

The authors declare no competing interests.
