## [Peer Review File · Nature Communications]

Tigerfish designs oligonucleotide-based *in situ* hybridization probes targeting intervals of highly repetitive DNA at the scale of genomesREVIEWER COMMENTS

Reviewer #1 (Remarks to the Author):

In this article, Aguilar and colleagues describe Tigerfish, a bioinformatic pipeline for the identification and design of oligonucleotide probes for repetitive DNA. Tigerfish can be run in different modes, which depend on the input, and yields sequences with specified parameters that can be synthesized as oligonucleotide probes and used for DNA FISH. The authors demonstrate successful use of Tigerfish by designing probes specific to the repetitive DNA sequences on each of the human chromosomes and using these probes on both mitotic chromosome spreads and in interphase nuclei. Overall, this manuscript is technically sound (although we lack the computational knowledge to run Tigerfish ourselves and cannot speak to the quality of the code). The major comment (elaborated below) is that it is somewhat unclear whether this new technique indeed fills a critical technical gap to provide novel insights. It is possible that the significance of this new technique is understated in this manuscript (in which case the author can elaborate on that, by textual changes) but as is it seems as if most (if not all) that Tigerfish achieves seems to be achieved by other existing techniques.

Major

- The significance of the Tigerfish bioinformatic pipeline is not explained. The authors state that a panel of probes could have utility in diagnostic and chromosomal enumeration assays, however they show that already existing probes designed through already existing pipelines can be used for this purpose and in fact use these probes on their mitotic spreads alongside the Tigerfish probes. Do Tigerfish probes have an advantage over these other probe sets? Additionally, methods already exist for de novo repeat identification and oligo optimization. Does Tigerfish have an advantage over these other pipelines and if so, can the authors elaborate on it?

- The authors claim that design of specific probes against repetitive regions is technically challenging, however repetitive regions have historically been significantly easier to design probes against. The repetitive nature of sequence limits the number of possible probe sequences, in addition, tools already exist for the identification of repeat sequences, and repeat optimization is not as critical as for non-repetitive sequences. Can the authors please elaborate on these technical challenges?

- The beauty of working with repetitive DNA is that the repetitive nature of the arrays/sequences provides natural signal amplification, greatly simplifying the DNA FISH procedure. From the details presented in the manuscript, it is not clear whether the probes designed using Tigerfish take advantage of these features of repetitive DNA. From the data presented in the manuscript, the copy number of the target sequences is unclear and the authors do not comment on the minimum number of copies needed or whether there is an optimum copy number so that experimental signal amplification becomes unnecessary.

Minor

- It is unclear why both metaphase chromosomes and interphase nuclei were used. The metaphase spreads are clear but in the interphase cells, the same number of foci is not always observed for Tigerfish and PaintSHOP probes. No comment is given for why the foci number is different with the two methods. Is this difference technical or biological? Can the authors please elaborate on what we can learn from looking at both metaphase chromosome spreads and interphase nuclei?

- In figure 3, for chromosomes 14, 21, and Y, it is hard to see both Tigerfish and PaintSHOP signals. Instead of showing both homologous chromosomes from the spread, it might be advantageous to show a split channel image so it's easier to see both probe sets.

- The method used for DNA FISH on interphase nuclei is missing.

- Could the authors comment on how Tigerfish handles short repeats? Many simple repeats are quite short - can Tigerfish use multiple copies as a probe? Would it ignore these repeats?

- Can the authors comment on whether the probe length and ideal T_m range are accommodating to the skewed AT/GC content of repeats?

- From the diagram in Figure 1, it is not clear how the workflow changes based on what "mode" Tigerfish is run in.

- In figure 2, consider switching the bars in panel b so that they match c-e with permissive on the left and conservative on the right.

- In the section "Computational requirements to run Tigerfish", in the fourth line and following the "1)", "he" should be "the"

Reviewer #2 (Remarks to the Author):

Fluorescence in situ hybridization (FISH) is a crucial technique used in molecular biology and genetics to visualize and localize specific DNA sequences within cells or tissues. To identify suitable oligonucleotide (oligo) probes for FISH experiments, numerous computational tools have been developed. However, because the repetitive sequences are frequent sources of unwanted background when performing FISH experiments, these methods were designed to avoid discovering candidate probes for repetitive sequences. On the other hand, repetitive DNA sequences also play important roles in many cellular and organismal functions and studying these regions via FISH could provide useful insights into these mechanisms.

The main contribution of this paper is to introduce a computational pipeline named Tigerfish for designing oligo probes targeting repetitive DNA regions. The authors also performed in situ experiments to validate the oligo probes designed by Tigerfish, and demonstrated the specificity and utility of Tigerfish for visualizing the repetitive DNA intervals in situ by metaphase and interphase FISH experiments. Overall, Tigerfish extends the toolkit of oligo-based FISH to highly repetitive DNA, providing new avenues for the functional studies of such sequences. The article is well written and the results are supported by data. However, the pipeline itself is not particularly novel.

Major comments:

1. My major concern is the novelty of the computational method proposed. Although it is true that it is particularly important to have a computational approach for designing oligo probes targeting repetitive DNA regions, the method introduced in this manuscript seems a simple integration of different existing tools, and thus lacking novelty in terms of methodology development.

2. The author claims that computational design of specific probes against repetitive DNA regions presents significant technical challenges but the evidence to support this claim is weak. Previous tools were designed to discover oligo probes for non-repetitive DNA regions, but it may be possible to easily adapt these tools for discovering probes targeting repetitive regions, for instance by disabling the filtering of probes containing highly abundant k-mers for tandem repeats. To strengthen the motivation behind this work, the authors should provide clear evidence of the technical challenges, or perform tests to demonstrate that previous tools are not effective in this regard.

3. In the paper, the authors primarily used the PaintSHOP for comparison with Tigerfish in both computational and experimental settings. Currently, there are many other oligo probe design software options available such as OligoArray, FISH Oracle, Stellaris Probe Designer, OligoFactory. How does the performance of Tigerfish compare with these design software programs? Considering the experimental workload, the authors can evaluate the accuracy of probes designed by different software in terms of performance metrics such as off-target rates.

4. Due to the varying characteristics of dispersed and tandem repetitive DNA sequences, the authors should design experiments to evaluate the performance of their proposed method in these two different scenarios.

5. In order to demonstrate the superiority of Tigerfish over previously designed computational tools, the authors should consider presenting experimental results or providing concrete examples of how Tigerfish overcomes the challenges encountered by these tools.

6. Please provide examples for discovering repetitive sequences with the Repeat Discovery Mode module and discovering oligo probes with the Probe Design Mode module, which could help readers understand the specific nature of this problem and the logic behind the proposed Tigerfish.

7. The authors designed genome-wide probe sets for 263 intervals, out of which 235 intervals were identified with the 'permissive' parameter settings, and only 28 intervals were identified by both parameter settings. What could be the possible reasons for this and whether there is a reasonable biological explanation? Furthermore, it would be interesting to know whether the authors have attempted to adjust the relevant parameters and observe if this proportion changes.

8. The texts in the 'Probe discovery at the scale of human genomes' section suggest that the predicted binding activities of the probes are depicted in Fig. 2d. However, the caption for Fig. 2d indicates that it shows the aggregate number of on-target binding predictions for probe sets, which is inconsistent with the description in the section. Moreover, the labels of Y-axes of Figs. 2d and 2e appear to be incorrect.

9. In the 'Computational requirements to run Tigerfish' section, the reported median runtime for all 263 intervals is 6.9 hours, whereas the median runtime for the group of intervals greater than 1 Mb is 4.6 hours. This result is puzzling, as the latter runtime should logically be longer than the former. This discrepancy may cause confusion.

10. The tool is valuable to the FISH-based assays, particularly for designing oligo probes targeting repetitive DNA regions. However, it would be helpful to discuss any limitations of Tigerfish and areas where it could be improved.

Minor comments:

1. In the 'Probe discovery at the scale of human genomes' section, the authors should use a consistent name for the assembly used. Specifically, both the human telomere-telomere CHM13v2 + HG002 chrY assembly and the human T2T CHM13v2 HG002 chrY assembly are mentioned in this section, and it would be better to use a unified name for clarity.

2. There are two duplicate sentences in the 'Competing Interest Statement' section.

3. What is the meaning of pink box in Fig. 3a?

4. Please carefully review the manuscript to confirm that all abbreviations have been appropriately defined, such as 'max_pdup_binding' in the 'Computing in silico binding predictions' section.

Reviewer #3 (Remarks to the Author):

Aguilar et al. present TigerFISH, a probe design method for DNA-FISH that can efficiently design probes for highly repetitive chromosomal regions. TigerFISH leverages previous packages, such as the fast k-mer counting algorithm Jellyfish, to speed up the probe design process for highly repetitive regions, making it possible (in terms of computational resources) to design probes for such regions on a genome-wide scale. The authors then validate TigerFISH for one repetitive region per chromosome via FISH experiments, and compare these results against non-repetitive probes (also one per chromosome) designed by PaintSHOP. The TigerFISH fluorescence shows up in the expected chromosomal locations, and the number of TigerFISH puncta shows good correspondence with the number of PaintSHOP puncta.

While the computational workflow behind TigerFISH is sound, I have some concerns related to its novelty, applicability, as well as broad interest in the field. Therefore, I do not recommend its publication in Nature Communications. However, this work can be published in a different journal if

the authors can address the following concerns.

Concerns related to novelty, general applicability, and broad interest

The first concern I have is related to the motivation for targeting highly repetitive regions, and why this is interesting to a broad audience. Previous FISH probe design programs usually skip these regions and instead focus on non-repetitive coding regions, which are more biologically interesting. The authors list several biological processes that involve repetitive regions of the chromosome, including the recruitment of the chromosome segregation machinery during mitosis, the protection of chromosome ends, and meiotic drive and speciation. However, the authors are not giving a compelling reason for why DNA FISH imaging these repetitive regions would shed light on these biological processes. For the verification and application of TigerFISH, the authors stain one repetitive region per chromosome and cite chromosome counting as one potential usage case. However, this is not a compelling enough reason, since there are numerous other ways for doing chromosome enumeration already. This paper can be strengthened if the authors can offer a more compelling reason for why DNA FISH imaging repetitive regions of the chromosome is interesting, and give some more detailed examples of how TigerFISH probes can shed light on specific biological processes

In addition, as the authors acknowledge, there exist previous methods for designing probes for repetitive regions of the chromosome (ref 45 ~ 57). The authors claim that these methods are not computationally scalable, and the computational efficiency of TigerFISH sets it apart from previous approaches. I wonder if the authors can add in more details regarding how unscalable these previous methods are, so that the novelty of TigerFISH can be strengthened

Concerns related to methodology and computational efficiency

In Figure 5, the authors show that TigerFISH can take up to 20 or more hours to design probes for some repetitive regions. Can the authors add in more details / insights on why this is the case, and / or potential ways to speed up the process for these regions?

Concerns related to data interpretation

In Figure 4c, is the puncta / cell histogram for all TigerFISH count data from all 46 chromosomes? If so, the authors should make this clear

In the same figure, it appears that some chromosomes in some nuclei have more than two counts by either TigerFISH probes, PaintSHOP probes, or both. Can the authors explain this? In addition, are there some chromosomes that are more likely to have more than two counts than others?

We thank the editor and editorial team for handling our manuscript and the reviewers for their thorough reading of the manuscript and constructive comments. Below, please find our responses to the reviewer comments and a summary of changes made to the manuscript.

REVIEWER COMMENTS

Reviewer #1 (Remarks to the Author):

In this article, Aguilar and colleagues describe Tigerfish, a bioinformatic pipeline for the identification and design of oligonucleotide probes for repetitive DNA. Tigerfish can be run in different modes, which depend on the input, and yields sequences with specified parameters that can be synthesized as oligonucleotide probes and used for DNA FISH. The authors demonstrate successful use of Tigerfish by designing probes specific to the repetitive DNA sequences on each of the human chromosomes and using these probes on both mitotic chromosome spreads and in interphase nuclei. Overall, this manuscript is technically sound (although we lack the computational knowledge to run Tigerfish ourselves and cannot speak to the quality of the code). The major comment (elaborated below) is that it is somewhat unclear whether this new technique indeed fills a critical technical gap to provide novel insights. It is possible that the significance of this new technique is understated in this manuscript (in which case the author can elaborate on that, by textual changes) but as is it seems as if most (if not all) that Tigerfish achieves seems to be achieved by other existing techniques.

Response: We thank the Reviewer for their careful reading and thoughtful critique of our manuscript. As described in detail below, we have added additional analyses and substantial textual changes to make the novelty and significance of Tigerfish more clear.

Major

- The significance of the Tigerfish bioinformatic pipeline is not explained. The authors state that a panel of probes could have utility in diagnostic and chromosomal enumeration assays, however they show that already existing probes designed through already existing pipelines can be used for this purpose and in fact use these probes on their mitotic spreads alongside the Tigerfish probes. Do Tigerfish probes have an advantage over these other probe sets? Additionally, methods already exist for de novo repeat identification and oligo optimization. Does Tigerfish have an advantage over these other pipelines and if so, can the authors elaborate on it?

Response: We thank the Reviewer for raising this point. In order to make the significance of Tigerfish more clear, we have added a new section to the **Results** entitled “**Challenges associated with designing probes that target repetitive DNA**” (page 5, line 10 – page 7, line 3). For brevity, the full text of this addition is not shown in this Response document. We also have added an accompanying set of new analyses as **Supplementary Figure 1**. Together, these additions more carefully and thoroughly explain the specific challenges encountered when designing probes against repetitive DNA, detail why existing tools cannot easily overcome these challenges, and provide a test case using human alpha satellite DNA repeats. We believe these additions will better contextualize the novelty of Tigerfish and provide quantitative support for our arguments.

- The authors claim that design of specific probes against repetitive regions is technically challenging, however repetitive regions have historically been significantly easier to design probes against. The

repetitive nature of sequence limits the number of possible probe sequences, in addition, tools already exist for the identification of repeat sequences, and repeat optimization is not as critical as for non-repetitive sequences. Can the authors please elaborate on these technical challenges?

Response: We thank the Reviewer for raising this point. We believe the addition of the new section to the **Results** entitled “**Challenges associated with designing probes that target repetitive DNA**” and the accompanying **Supplementary Figure 1** satisfies this request.

- The beauty of working with repetitive DNA is that the repetitive nature of the arrays/sequences provides natural signal amplification, greatly simplifying the DNA FISH procedure. From the details presented in the manuscript, it is not clear whether the probes designed using Tigerfish take advantage of these features of repetitive DNA. From the data presented in the manuscript, the copy number of the target sequences is unclear and the authors do not comment on the minimum number of copies needed or whether there is an optimum copy number so that experimental signal amplification becomes unnecessary.

Response: We thank the Reviewer for raising these points. We have addressed them via the addition of the following text to the **Results** (page 9, line 22, new text in blue font):

“When factoring in the size of the target intervals, we observed target site densities of 0.017–798.6 target sites per kb (median: 47.9 target sites per kb) for the ‘permissive’ group and 0.64–475.9 target sites per kb (median: 6.4 target sites per kb) for the ‘conservative’ group (**Fig. 2e**). Thus, the majority of the probe sets meet if not greatly exceed the threshold of ~200 target sites that in our experience leads to reliably robust DNA FISH; importantly, as this threshold is enforced by the efficiency of probe hybridization rather than absolute signal strength, we have observed that >200 probes is optimal even if signal amplification approaches such as Rolling Circle Amplification (RCA)⁶⁹, Hybridization Chain Reaction (HCR)⁷⁰, or Signal Amplification by Exchange Reaction (SABER)⁷¹ are employed. Moreover, due to the reiterated nature of the target intervals, hundreds to many of thousands of target sites can be labeled by just one or a few probes, greatly reducing the cost of the FISH relative to experiments that target non-repetitive DNA with sets of hundreds to thousands of oligo probes. Thus, while Tigerfish is theoretically capable of designing probes against input intervals of any size, we recommend targeting intervals greater than 10 kb in length to maximize the chance of experimental success by providing >200 binding sites for non-overlapping 40–50 nt probes; such regions are mostly to occur as tandem repeats and appear less frequently as interspersed repeats (**Fig. 2b**).”

Minor

- It is unclear why both metaphase chromosomes and interphase nuclei were used. The metaphase spreads are clear but in the interphase cells, the same number of foci is not always observed for Tigerfish and PaintSHOP probes. No comment is given for why the foci number is different with the two methods. Is this difference technical or biological? Can the authors please elaborate on what we can learn from looking at both metaphase chromosome spreads and interphase nuclei?

Response: We thank the Reviewer for raising this point. In order to address it, we have introduced new text to the **Results** section explaining the rationale behind including the interphase data and to provide some potential explanations for the observed differences in the number of detected puncta between the Tigerfish and PaintSHOP probe sets (page 10, line 22, new text in blue font):

“In order to augment our metaphase data with a sample type that provides a means to visually inspect that we were achieving the correct staining pattern with a larger sample size, we also performed a series of 24 interphase FISH experiments on 46,XY primary human lymphoblasts using the same Tigerfish and PaintSHOP probe set combinations as a means to visually enumerate chromosomal copy number (**Fig. 4a**). Specifically, we imaged >40 cells for each experiment and quantified the number of observed Tigerfish and PaintSHOP foci in the 3D volume of the nucleus (**Fig. 4b, Supplementary Fig. 7–10**). Our analysis of the resulting data revealed a strong agreement between the two types of probe set (78.4% concordance, $n = 1,061$), with both approaches predominantly displaying 2 foci per nucleus (PaintSHOP: 781/1061, 73.6%; Tigerfish: 922/1061, 86.9%) and identifying a range of foci (1–4) per nucleus consistent with our previous studies using oligo-based probes for enumeration^{10,22,72} (**Fig. 4c**). We also noted that we consistently identified more Tigerfish foci than PaintSHOP foci. Although we cannot formally rule out differences in the underlying copy number of the target loci or frequency of sister chromatid separation as the two probe set types target distinct chromosomal regions, we believe the most likely explanation for this observation is that the PaintSHOP probes required three rounds of hybridization (primary oligo library, secondary PER-extended “bridge” oligos, tertiary fluorescently labeled “imager” oligos), while the Tigerfish probes only required two (primary PER-extended probe oligos, secondary imager oligos). We and others have observed reduced labeling efficiency due to the additional hybridization round required to use bridge oligos^{12–14,73}, and further support for this observation comes from the higher frequency of nuclei with no detectable PaintSHOP foci relative to the number with no detectable Tigerfish foci (Tigerfish: 13/1061, 1.2%; PaintSHOP: 48/1061, 4.5%; two-sided Fisher’s exact p -value = 5.7×10^{-6}).”

- In figure 3, for chromosomes 14, 21, and Y, it is hard to see both Tigerfish and PaintSHOP signals. Instead of showing both homologous chromosomes from the spread, it might be advantageous to show a split channel image so it's easier to see both probe sets.

Response: We thank the Reviewer for raising this point. We have added arrow call-outs to the relevant **Figure 3** panels to help guide readers to the location of the foci. We feel that there is value to showing both alleles in our spread data and adding split channel images on top of the existing data would make the figure too busy but are open to adding the split channel images as a separate SI figure if the arrow call-outs are not sufficient.

- The method used for DNA FISH on interphase nuclei is missing.

Response: We thank the Reviewer for raising this point and apologize for this omission. The protocol for the interphase FISH is identical to that used for metaphase FISH as the metaphase spread samples also have quite a lot of intact nuclei on them. We have added clarifying text to the appropriate place in the **Methods** section.

- Could the authors comment on how Tigerfish handles short repeats? Many simple repeats are quite short - can Tigerfish use multiple copies as a probe? Would it ignore these repeats?

Response: We thank the Reviewer for raising this point. We agree readers would benefit from an expanded discussion about the constraints imposed by the target repeat length. We have added text accordingly to the **Results** section (page 9, line 34, next text in blue font):

“Thus, while Tigerfish is theoretically capable of designing probes against input intervals of any size, we recommend targeting intervals greater than 10 kb in length to maximize the chance of experimental success by providing >200 binding sites for non-overlapping 40–50 nt probes; such regions are mostly to occur as tandem repeats and appear less frequently as interspersed repeats (**Fig. 2b**). “

- Can the authors comment on whether the probe length and ideal T_m range are accommodating to the skewed AT/GC content of repeats?

Response: We thank the Reviewer for raising this question. Tigerfish can be set to run with relatively broad bounds on the allowed T_m and length of probes its designs, which in turn enables it design probes against targets with different AT/GC content skews in parallel. To make this more clear to the readers, we have added the additional panels **f** and **g** to **Figure 2** and introduced the accompanying text in the **Results** section (page 9, line 38, new text in blue font):

“Thus, while Tigerfish is theoretically capable of designing probes against input intervals of any size, we recommend targeting intervals greater than 10 kb in length to maximize the chance of experimental success by providing >200 binding sites for non-overlapping 40–50 nt probes; such regions are mostly to occur as tandem repeats and appear less frequently as interspersed repeats (**Fig. 2b**). On the level of the probes themselves, as we allowed a broad range of permissible lengths (25–50 nt), %G+C content (20–80%), and formamide-adjusting melting temperatures (42–52°C), we observed a broad distribution of probe lengths (**Fig. 2f**) and formamide-adjusted melting temperatures (**Fig. 2g**), indicating that when Tigerfish is run with flexible probe parameter choices genome-wide it is able to design probes suitable for the variable nucleotide compositions found at its repetitive targets.”

- From the diagram in Figure 1, it is not clear how the workflow changes based on what "mode" Tigerfish is run in.

Response: We thank the Reviewer for raising this point and apologize that the relationship between the different “modes” of Tigerfish was not clear. The three modes of Tigerfish simply represent three distinct entry points into what is otherwise a collection of identical steps. In order to make this more clear to readers, we have added additional graphical elements to the schematic in **Figure 1** to indicate where each mode enters. We also have introduced clarifying text into the **Results** section of the manuscript (page 7, line 19, new text in blue font):

“Tigerfish can be run in one of three execution modes; these modes do not differ in the logic they use for designing and evaluating probes but allow different entry points into the process depending on what information the researcher already has in hand (**Fig. 1**). The first, termed “Repeat Discovery Mode”, runs the full Tigerfish workflow end to end and is intended for cases in which researchers do not have a *priori* knowledge of where repetitive regions occur in their genome of interest. In Repeat Discover Mode, users list genomic scaffolds where *de novo* repeat discovery and probe design is to be performed in the configuration file. Repeat Discovery Mode uses a *k*-mer counting strategy to identify repetitive DNA regions *de novo* by identifying intervals that contain *k*-mers with high abundance in the genome (**Methods**). Users can tune the size of the search window and the magnitude of the *k*-mer count values needed for an interval to be flagged as repetitive, thereby controlling the nature of the repeat regions identified. The second, termed “Probe Design Mode”, skips repeat discovery step and runs the

Tigerfish pipeline starting from the probe design step (Fig. 1). Probe Design Mode is intended for instances where the genomic interval(s) a user wants to target for probe design are already known. In this case, the user must provide an additional BED-formatted file⁶⁸ that specifies the genomic coordinates for interval(s) to perform probe design against. The third, termed “Probe Analysis Mode”, runs the pipeline starting at the specificity filtering step that comes downstream of probe design (Fig. 1). Probe Analysis Mode is provided as a way to generate a new set of *in silico* binding predictions for probes contained in an existing Tigerfish output file; this functionality may be used to predict how the binding pattern of the input probes might change as the result of altering the salt concentration or melting temperature of the hybridization reaction.”

- In figure 2, consider switching the bars in panel b so that they match c-e with permissive on the left and conservative on the right.

Response: We thank the Reviewer for this helpful suggestion. We have made this change.

- In the section "Computational requirements to run Tigerfish", in the fourth line and following the "1)", "he" should be "the"

Response: We thank the Reviewer for catching this error. We have fixed it.

Reviewer #2 (Remarks to the Author):

Fluorescence in situ hybridization (FISH) is a crucial technique used in molecular biology and genetics to visualize and localize specific DNA sequences within cells or tissues. To identify suitable oligonucleotide (oligo) probes for FISH experiments, numerous computational tools have been developed. However, because the repetitive sequences are frequent sources of unwanted background when performing FISH experiments, these methods were designed to avoid discovering candidate probes for repetitive sequences. On the other hand, repetitive DNA sequences also play important roles in many cellular and organismal functions and studying these regions via FISH could provide useful insights into these mechanisms.

The main contribution of this paper is to introduce a computational pipeline named Tigerfish for designing oligo probes targeting repetitive DNA regions. The authors also performed in situ experiments to validate the oligo probes designed by Tigerfish, and demonstrated the specificity and utility of Tigerfish for visualizing the repetitive DNA intervals in situ by metaphase and interphase FISH experiments. Overall, Tigerfish extends the toolkit of oligo-based FISH to highly repetitive DNA, providing new avenues for the functional studies of such sequences. The article is well written and the results are supported by data. However, the pipeline itself is not particularly novel.

Response: We thank the Reviewer for their careful evaluation of our manuscript and kind words. As described in detail below (and as reiterated from responses to Reviewer #1 above), we have added additional analyses and substantial textual changes to make the novelty and significance of Tigerfish more clear.

Major comments:

1. My major concern is the novelty of the computational method proposed. Although it is true that it is particularly important to have a computational approach for designing oligo probes targeting repetitive DNA regions, the method introduced in this manuscript seems a simple integration of different existing tools, and thus lacking novelty in terms of methodology development.

Response: We thank the Reviewer for raising this point. In order to make the significance of Tigerfish more clear, we have added a new section to the **Results** entitled “**Challenges associated with designing probes that target repetitive DNA**” (page 5, line 10 – page 7, line 3). For brevity, the full text of this addition is not shown in this Response document. We also have added an accompanying set of new analyses as **Supplementary Figure 1**. Together, these additions more carefully and thoroughly explain the specific challenges encountered when designing probes against repetitive DNA, detail why existing tools cannot easily overcome these challenges, and provide a test case using human alpha satellite DNA repeats. We believe these additions will better contextualize the novelty of Tigerfish and provide quantitative support for our arguments.

2. The author claims that computational design of specific probes against repetitive DNA regions presents significant technical challenges but the evidence to support this claim is weak. Previous tools were designed to discover oligo probes for non-repetitive DNA regions, but it may be possible to easily adapt these tools for discovering probes targeting repetitive regions, for instance by disabling the filtering of probes containing highly abundant k-mers for tandem repeats. To strengthen the motivation behind this work, the authors should provide clear evidence of the technical challenges, or perform tests to demonstrate that previous tools are not effective in this regard.

Response: We thank the Reviewer for raising this point. We believe the addition of the new section to the **Results** entitled “**Challenges associated with designing probes that target repetitive DNA**” and the accompanying **Supplementary Figure 1** satisfies this request.

3. In the paper, the authors primarily used the PaintSHOP for comparison with Tigerfish in both computational and experimental settings. Currently, there are many other oligo probe design software options available such as OligoArray, FISH Oracle, Stellaris Probe Designer, OligoFactory. How does the performance of Tigerfish compare with these design software programs? Considering the experimental workload, the authors can evaluate the accuracy of probes designed by different software in terms of performance metrics such as off-target rates.

Response: We thank the Reviewer for raising this point. The novelty of Tigerfish does not rest at the level of selecting probe candidates from input sequences, which is where the software tools raised primarily operate. Instead, the significance and novelty of Tigerfish lies in the processes by which it evaluates the on- and off-target specificity for a given probe candidate given a target interval. To make this distinction more clear, we have added the following text to the **Results** section (page 5, line 29, new text in blue font):

“Existing computational probe design methods attempt to avoid designing non-specific probes using one or more of the follow three main approaches: 1.) Using existing genome annotations (e.g., from repeatMasker³⁴) to identify intervals of repetitive DNA and prohibiting design from these intervals (Stellaris Probe Designer, Chorus²⁰); 2.) Aligning efficacious probes outputted by upstream steps to the reference genome and discarding those aligning more than one or a few times (OligoArray¹⁸, OligoMiner²², ProbeDealer²⁴, Chorus²⁰, Chorus2²⁵); 3.) Filtering efficacious

probes outputted by upstream that contain the presence of k -mers with many occurrences in the reference genome (OligoMiner²², iFISH²³, PROBER¹⁹, Chorus2²⁵). Collectively, these approaches are purpose-built to avoid designing probes that contain stretches of highly repetitive DNA, as such probes are an extremely problematic source of off-target background when the intended target is an interval of non-repetitive DNA^{31,32}. However, if intervals of repetitive DNA are instead the intended design target, it is difficult or impossible to adopt existing workflows because of these filtering approaches, which are almost always hard-coded into the probe design tools. Specifically, approach 1 precludes the design of probes against any annotated repetitive region and thus cannot be present in any pipeline seeking to design probes against repetitive DNA intervals. Approach 2 is likewise problematic, as probes that target repetitive DNA by definition occur in the genome many times and thus would return many distinct alignment sites, but most probe design tools purposely filter any probe with more than 1 high-scoring alignment site. Finally, while approach 3 can be tweaked in some tools (e.g., OligoMiner²²) to allow probes containing high-occurrence k -mers to be designed, they do not contain any computational infrastructure to allow users to determine what proportion the occurrences of the relevant k -mers derive from the target interval; occurrences deriving from other genomic locations could lead to unacceptable levels of off-target background.”

4. Due to the varying characteristics of dispersed and tandem repetitive DNA sequences, the authors should design experiments to evaluate the performance of their proposed method in these two different scenarios.

Response: We thank the Reviewer for raising this point. We do not anticipate that Tigerfish will perform differently in terms of probe design or experimental results due to a target being a tandem or dispersed/interspersed repeat. To make this clear to readers we have:

1. Remade **Figure 2b** to such that the number and proportion of the overlapping RepeatMasker annotations with targets discovered in the genome-scale Tigerfish runs are labeled as being interspersed or tandem.
2. Added information about the RepeatMasker Annotation (if known) of each of the 24 probe sets used for experimental validation to **Table 1**.
3. Added the following text to the **Results** section (page 9, line 34, new text in blue font):

“Thus, while Tigerfish is theoretically capable of designing probes against input intervals of any size, we recommend targeting intervals greater than 10 kb in length to maximize the chance of experimental success by providing >200 binding sites for non-overlapping 40–50 nt probes; such regions are mostly to occur as tandem repeats and appear less frequently as interspersed repeats (**Fig. 2b**).”

5. In order to demonstrate the superiority of Tigerfish over previously designed computational tools, the authors should consider presenting experimental results or providing concrete examples of how Tigerfish overcomes the challenges encountered by these tools.

Response: We thank the Reviewer for raising this point. We believe the addition of the new section to the **Results** entitled “**Challenges associated with designing probes that target repetitive DNA**” and the accompanying **Supplementary Figure 1** satisfies this request.

6. Please provide examples for discovering repetitive sequences with the Repeat Discovery Mode module and discovering oligo probes with the Probe Design Mode module, which could help readers understand the specific nature of this problem and the logic behind the proposed Tigerfish.

Response: We thank the Reviewer for raising this point. The three modes of Tigerfish simply represent three distinct entry points into what is otherwise a collection of identical steps. In order to make this more clear to readers, we have added additional graphical elements to the schematic in **Figure 1** to indicate where each mode enters. We also have introduced clarifying text into the **Results** section of the manuscript (page 7, line 19, new text in blue font):

“Tigerfish can be run in one of three execution modes; these modes do not differ in the logic they use for designing and evaluating probes but allow different entry points into the process depending on what information the researcher already has in hand (**Fig. 1**). The first, termed “Repeat Discovery Mode”, runs the full Tigerfish workflow end to end and is intended for cases in which researchers do not have *a priori* knowledge of where repetitive regions occur in their genome of interest. In Repeat Discover Mode, users list genomic scaffolds where *de novo* repeat discovery and probe design is to be performed in the configuration file. Repeat Discovery Mode uses a *k*-mer counting strategy to identify repetitive DNA regions *de novo* by identifying intervals that contain *k*-mers with high abundance in the genome (**Methods**). Users can tune the size of the search window and the magnitude of the *k*-mer count values needed for an interval to be flagged as repetitive, thereby controlling the nature of the repeat regions identified. The second, termed “Probe Design Mode”, skips repeat discovery step and runs the Tigerfish pipeline starting from the probe design step (**Fig. 1**). Probe Design Mode is intended for instances where the genomic interval(s) a user wants to target for probe design are already known. In this case, the user must provide an additional BED-formatted file⁶⁸ that specifies the genomic coordinates for interval(s) to perform probe design against. The third, termed “Probe Analysis Mode”, runs the pipeline starting at the specificity filtering step that comes downstream of probe design (**Fig. 1**). Probe Analysis Mode is provided as a way to generate a new set of *in silico* binding predictions for probes contained in an existing Tigerfish output file; this functionality may be used to predict how the binding pattern of the input probes might change as the result of altering the salt concentration or melting temperature of the hybridization reaction.”

7. The authors designed genome-wide probe sets for 263 intervals, out of which 235 intervals were identified with the 'permissive' parameter settings, and only 28 intervals were identified by both parameter settings. What could be the possible reasons for this and whether there is a reasonable biological explanation? Furthermore, it would be interesting to know whether the authors have attempted to adjust the relevant parameters and observe if this proportion changes.

Response: We thank the Reviewer for raising this point. We have added additional discussion of these findings in the **Results** section (page 8, line 38, new text in blue font):

“In order to showcase how users can tune parameters to optimize their design for different types of repeat regions, we performed our genome-scale runs with two sets of parameter groupings: 1) a ‘conservative’ set that prioritizes identifying large intervals of highly repetitive sequence such as those found at pericentromeres in order to prioritize extremely robust probes (>500 target sites); 2) a ‘permissive’ set that aims to exhaustively discover intervals of repetitive DNA that can be probed with >25 target sites (**Supplementary Data 1**). Analysis of

our genome-wide probe design runs revealed that Tigerfish was able to design probe sets for 28 intervals using the 'conservative' parameter set (**Supplementary Data 2**) and 263 intervals using the 'permissive' parameter set (**Supplementary Data 3**). As neither parameter set puts an upper bound on the number of target sites or the size of the target interval, the 28 intervals discovered using 'conservative' parameter settings were also present discovered using the 'permissive' parameter settings.”

While we anticipate that changing the parameters used to run Tigerfish would result in differences in the targets and probes discovered, we feel the current datasets adequately demonstrate the ability to perform genome-scale design with Tigerfish and adding additional genome-scale runs would not alter the overall conclusion. We agree that seeing how parameter changes influences what is returned is an interesting question, but believe such analyses are beyond the scope of this manuscript, which is focused on describing a new tool and how it works.

8. The texts in the 'Probe discovery at the scale of human genomes' section suggest that the predicted binding activities of the probes are depicted in Fig. 2d. However, the caption for Fig. 2d indicates that it shows the aggregate number of on-target binding predictions for probe sets, which is inconsistent with the description in the section. Moreover, the labels of Y-axes of Figs. 2d and 2e appear to be incorrect.

Response: We thank the Reviewer for raising this point. We apologize that what is being depicted in **Figure 2d** was not clear. To clarify, we added the following text to the **Results** section (page 9, line 19, new text in blue font):

“Our *in silico* specificity profiling also revealed a broad distribution of the aggregate number of predicted on-target binding sites for the probes or probe sets covering the 263 intervals, ranging from 25–30,972 target sites in the 'permissive' group (median: 236.9 target sites) and 500–30,972 targets sites in the 'conservative group (median: 20,165.2 target sites) (**Fig. 2d**).”

The Y-axis label of **Figure 2e** was indeed incorrect. We apologize and thank the Reviewer for catching this. The panel has been updated with the correct label.

9. In the 'Computational requirements to run Tigerfish' section, the reported median runtime for all 263 intervals is 6.9 hours, whereas the median runtime for the group of intervals greater than 1 Mb is 4.6 hours. This result is puzzling, as the latter runtime should logically be longer than the former. This discrepancy may cause confusion.

Response: We thank the Reviewer for raising this point. We apologize, that text was erroneous, the median runtime for all 263 intervals was 6.9 *minutes*. We thank the Reviewer for catching this regrettable typo and have updated the corresponding text in the **Results** section with the correct units (page 11, line 16, new text in blue font):

“Our analyses revealed that probe design against the majority of target intervals finished quickly, with a median run time of 6.9 *minutes* (range: 1.8 min – 50.2 hr) and a median CPU time of 5.2 *minutes* (range: 0.6 min – 4.9 hr) (**Fig. 5a,b**).”

10. The tool is valuable to the FISH-based assays, particularly for designing oligo probes targeting repetitive DNA regions. However, it would be helpful to discuss any limitations of Tigerfish and areas where it could be improved.

Response: We thank the Reviewer for this great suggestion. We have added the following text to the **Discussion** section (page 12, line 19, new text in blue font):

“Like all tools, Tigerfish has limitations and avenues for future development. For instance, the command-line nature of Tigerfish and its optimization for cluster-based computing may prove to be a barrier to entry for some users. Future work to implement a graphical user interface and to provide a cloud-based platform for running Tigerfish may help to make it accessible to a broader set of researchers. Tigerfish’s ability to design probes against specific intervals of highly repetitive DNA is also highly dependent on the quality of input genome assembly, and to date there are only a few fully polished telomere to telomere assemblies optimal for Tigerfish deployment. Nevertheless, we anticipate Tigerfish will play a key role in the experimental validation and biological investigation of repetitive DNA intervals as more fully assembled human, vertebrate, plant, and other model organism genomes continue to be introduced.”

Minor comments:

1. In the 'Probe discovery at the scale of human genomes' section, the authors should use a consistent name for the assembly used. Specifically, both the human telomere-telomere CHM13v2 + HG002 chrY assembly and the human T2T CHM13v2 HG002 chrY assembly are mentioned in this section, and it would be better to use a unified name for clarity.

Response: We thank the Reviewer for this helpful suggestion. We have unified the description of this genome assembly in the manuscript as “the T2T CHM13v2 + HG002 chrY assembly”.

2. There are two duplicate sentences in the 'Competing Interest Statement' section.

Response: We thank the Reviewer for catching this. The duplicate sentence has been removed.

3. What is the meaning of pink box in Fig. 3a?

Response: We thank the Reviewer for raising this point and apologize it was unclear. The pink box was meant to signify a microscope field of view. For simplicity, we have removed it.

4. Please carefully review the manuscript to confirm that all abbreviations have been appropriately defined, such as 'max_pdups_binding' in the 'Computing in silico binding predictions' section.

Response: We thank the Reviewer for raising this point. We have done this and added a definition to 'max_pdups_binding' in the relevant section.

Reviewer #3 (Remarks to the Author):

Aguilar et al. present TigerFISH, a probe design method for DNA-FISH that can efficiently design probes for highly repetitive chromosomal regions. TigerFISH leverages previous packages, such as the fast k-mer counting algorithm Jellyfish, to speed up the probe design process for highly repetitive regions, making it possible (in terms of computational resources) to design probes for such regions on a genome-wide scale. The authors then validate TigerFISH for one repetitive region per chromosome via FISH experiments, and compare these results against non-repetitive probes (also one per chromosome) designed by PaintSHOP. The TigerFISH fluorescence shows up in the expected chromosomal locations, and the number of TigerFISH puncta shows good correspondence with the number of PaintSHOP puncta.

While the computational workflow behind TigerFISH is sound, I have some concerns related to its novelty, applicability, as well as broad interest in the field. Therefore, I do not recommend its publication in Nature Communications. However, this work can be published in a different journal if the authors can address the following concerns.

Response: We thank the Reviewer for taking the time to read our manuscript and provide constructive feedback and a frank assessment of our work. We feel that the additions made to the manuscript in this resubmission help to clarify the significance, novelty, and potential applications of Tigerfish.

Concerns related to novelty, general applicability, and broad interest

The first concern I have is related to the motivation for targeting highly repetitive regions, and why this is interesting to a broad audience. Previous FISH probe design programs usually skip these regions and instead focus on non-repetitive coding regions, which are more biologically interesting. The authors list several biological processes that involve repetitive regions of the chromosome, including the recruitment of the chromosome segregation machinery during mitosis, the protection of chromosome ends, and meiotic drive and speciation. However, the authors are not giving a compelling reason for why DNA FISH imaging these repetitive regions would shed light on these biological processes. For the verification and application of TigerFISH, the authors stain one repetitive region per chromosome and cite chromosome counting as one potential usage case. However, this is not a compelling enough reason, since there are numerous other ways for doing chromosome enumeration already. This paper can be strengthened if the authors can offer a more compelling reason for why DNA FISH imaging repetitive regions of the chromosome is interesting, and give some more detailed examples of how TigerFISH probes can shed light on specific biological processes

Response: We thank the Reviewer for raising this point. In order to address it, we have added additional text to the **Introduction** section highlighting potential biological processes that Tigerfish probes could help investigate (page 4, line 14, new text in blue font):

“Thus, more detailed studies of highly repetitive DNA regions and their transcription products through **low-cost** targeted assays such as FISH may help uncover the mechanisms by which these mysterious regions exert their influence on important biological processes. **For instance, the targeted visualization of repetitive regions would allow the assessment of chromatin compaction at the single cell level⁴³, the quantification of mitotic errors such as anaphase bridges⁴⁴, and the investigation of the micronucleation frequency of a given element⁴⁵; as repetitive DNA regions are frequent sources of mitotic errors⁴⁶, such experiments may help define the mechanisms by which genome stability is maintained.**”

In addition, as the authors acknowledge, there exist previous methods for designing probes for repetitive regions of the chromosome (ref 45 ~ 57). The authors claim that these methods are not computationally scalable, and the computational efficiency of TigerFISH sets it apart from previous approaches. I wonder if the authors can add in more details regarding how unscalable these previous methods are, so that the novelty of TigerFISH can be strengthened

Response: We thank the Reviewer for raising this point. We believe the addition of the new section to the **Results** entitled “**Challenges associated with designing probes that target repetitive DNA**” and the accompanying **Supplementary Figure 1** satisfies this request.

Concerns related to methodology and computational efficiency

In Figure 5, the authors show that TigerFISH can take up to 20 or more hours to design probes for some repetitive regions. Can the authors add in more details / insights on why this is the case, and / or potential ways to speed up the process for these regions?

Response: We thank the Reviewer for raising this point. We have added the following text to the **Results** section to provide additional information about factors that influence the run time of Tigerfish (page 11, line 31, new text in blue font):

“As large genomic targets often will contain many possible candidate probes, it may take Tigerfish some time to arrive on one or more suitable probes that satisfy the user-specified requirements for the aggregate on-target binding while only including probes with low predicted off-target binding activity at such targets. While 1–2 day compute runs are often standard for genome-scale probe design tasks in large and complex genomes^{22,26}, users may still attempt to reduce the processing time for problematic intervals by adjusting the parameters used for the rank-ordering of candidate probes or relaxing the probe design requirements (**Supplementary Note**).”

Concerns related to data interpretation

In Figure 4c, is the puncta / cell histogram for all TigerFISH count data from all 46 chromosomes? If so, the authors should make this clear

Response: We thank the Reviewer for raising this point and apologize that it was unclear. The puncta from all 24 chromosomes targeted are indeed shown in aggregate in **Figure 4c**. We have added a panel label to make this clear for readers.

In the same figure, it appears that some chromosomes in some nuclei have more than two counts by either TigerFISH probes, PaintSHOP probes, or both. Can the authors explain this? In addition, are there some chromosomes that are more likely to have more than two counts than others?

Response: We thank the Reviewer for raising this point. Nuclei with three and four foci are frequently observed when performing enumeration experiments and most likely contain replicated sister chromatids that have separated. We have added a note to the legend for **Figure 4** with this information. We also examined the frequencies of having >2 puncta in the Tigerfish and PaintSHOP datasets and did not see individual chromosomes being significantly

more likely to display this phenotype; the distribution was largely uniform. For simplicity, we'd prefer to leave this analysis out of the manuscript.

We have also made several minor changes to the manuscript in the process of addressing the concerns raised by the Reviewers. All textual additions are marked with blue font. The most substantive of these changes are listed below:

1. As we have added a new **Supplementary Figure 1**, the title of all existing supplementary figures has been incremented by one to accommodate (e.g., **Supplementary Figure 2** in the original submission is now **Supplementary Figure 3**).
2. We have added 13 references (marked with blue font in the updated manuscript document).
3. As a result of some of the textual additions, we have reordered the supplementary data files:

Resubmission	Original submission
Supplementary Data 1	Supplementary Data 5
Supplementary Data 2	Supplementary Data 2
Supplementary Data 3	Supplementary Data 1
Supplementary Data 4	Supplementary Data 3
Supplementary Data 5	Supplementary Data 4

REVIEWERS' COMMENTS

Reviewer #1 (Remarks to the Author):

The authors have addressed most of concerns raised in the first round of the review. It seems that all the reviewers raised the same 'high level' concerns regarding the novelty of the work in the first round. The authors expanded the result section to introduce the issues (challenges, the need of TigerFish), which made the manuscript much stronger. I remain agnostic about how novel is novel enough, as I believe it must be a decision made by the editorial office. The manuscript is technically sound, and achieves what they intended to achieve. Overall, this is a solid paper, and as far as there is a consensus that this work is of sufficient interest to a sufficiently broad audience, I have no objection publishing this paper.

Reviewer #2 (Remarks to the Author):

I appreciate the authors' efforts in addressing my previous comments. Although they did not perfectly address my comments regarding to the novelty of the proposed method, I am fine with the acceptance of the current version.

Reviewer #3 (Remarks to the Author):

We thank the authors for addressing our concerns related to methodology and data interpretation. I believe the authors' response to our concerns on methodology and data interpretation is sufficient. Overall, this paper presents a bioinformatics package that is able to design probes for highly repetitive DNA regions on the genome scale. The paper is well-written and the results are well-validated.

Nevertheless, I have lingering concerns regarding the novelty of the TigerFISH bioinformatics package in both the originality of the code as well as the novelty of its area of application.

In terms of the novelty of the code, TigerFISH seems to widely utilize code from other already published bioinformatics packages (including Jellyfish, OligoMiner, Bowtie2, and SAMtools) to support core parts of its function, such as k-mer counting and oligo probe design. While it can be argued that TigerFISH pieces these other packages together in a streamlined fashion for the purpose of designing probes for highly repetitive DNA regions, in this light TigerFISH is best presented as an application case of these other bioinformatics packages for addressing one specific biological question, rather than a novel bioinformatics package in and of itself.

In terms of the novelty of the use case, as the authors acknowledge it is already possible to design FISH probes for highly repetitive DNA regions with existing bioinformatics packages such as OligoMiner + PaintSHOP. The authors benchmarked the performance of this combination on human chromosome-specific alpha satellite arrays, on either the full array or consensus monomers discovered by Tandem Repeat Finder. While it is true that it takes too long to filter probes out for specificity if all probes discovered on the full array are considered, the CPU resource usage for discovering and then filtering probes from consensus monomers is quite reasonable (1 ~ 2 h for the four examples the authors showed). Comparing this to the CPU resource usage of TigerFISH in Figure 5, where ~ 90 % of repetitive DNA regions also take 1 ~ 2 CPU hours to design probes for, the resource usage of the already published OligoMiner + PaintSHOP combination seems to fall within this range as well. Therefore, based on the four chromosome-specific alpha satellite array examples the authors provided, it is hard to establish that TigerFISH indeed outperforms previous methods in computational efficiency for the purpose of designing probes for highly repetitive DNA regions.

We thank the editor and editorial team for handling our manuscript and the reviewers for their thorough reading of the manuscript and constructive comments. Below, please find our responses to the reviewer comments and a summary of changes made to the manuscript.

REVIEWER COMMENTS

Reviewer #1 (Remarks to the Author):

The authors have addressed most of concerns raised in the first round of the review. It seems that all the reviewers raised the same 'high level' concerns regarding the novelty of the work in the first round. The authors expanded the result section to introduce the issues (challenges, the need of TigerFish), which made the manuscript much stronger. I remain agnostic about how novel is novel enough, as I believe it must be a decision made by the editorial office. The manuscript is technically sound, and achieves what they intended to achieve. Overall, this is a solid paper, and as far as there is a consensus that this work is of sufficient interest to a sufficiently broad audience, I have no objection publishing this paper.

Response: We thank the Reviewer for their careful reading of our revised manuscript and supportive comments.

Reviewer #2 (Remarks to the Author):

I appreciate the authors' efforts in addressing my previous comments. Although they did not perfectly address my comments regarding to the novelty of the proposed method, I am fine with the acceptance of the current version.

Response: We thank the Reviewer for their careful reading of our revised manuscript and direct feedback.

Reviewer #3 (Remarks to the Author):

We thank the authors for addressing our concerns related to methodology and data interpretation. I believe the authors' response to our concerns on methodology and data interpretation is sufficient. Overall, this paper presents a bioinformatics package that is able to design probes for highly repetitive DNA regions on the genome scale. The paper is well-written and the results are well-validated.

Response: We thank the Reviewer for their careful reading of our revised manuscript and supportive comments.

Nevertheless, I have lingering concerns regarding the novelty of the TigerFISH bioinformatics package in both the originality of the code as well as the novelty of its area of application.

In terms of the novelty of the code, TigerFISH seems to widely utilize code from other already published bioinformatics packages (including Jellyfish, OligoMiner, Bowtie2, and SAMtools) to support core parts of its function, such as k-mer counting and oligo probe design. While it can be argued that TigerFISH pieces these

other packages together in a streamlined fashion for the purpose of designing probes for highly repetitive DNA regions, in this light TigerFISH is best presented as an application case of these other bioinformatics packages for addressing one specific biological question, rather than a novel bioinformatics package in and of itself.

Response: We thank the Reviewer for raising this point. Tigerfish (similar to OligoMiner, PaintSHOP, Chorus2, ProbeDealer, and other computational tools) indeed relies on several existing bioinformatic packages, as these packages are field standards for achieving common computational tasks involved in the probe design workflow and are already highly optimized. However, Tigerfish does also introduce novel functionalities including a lightweight repeat region discovery algorithm and the *in silico* simulation framework to predict the interval-specific on- and off-target binding properties of probes containing highly reiterated DNA sequences.

In order to clarify the nature of Tigerfish with respect to existing bioinformatic packages, we have added the following text to the **Results** section (page 7, line 5, new text in blue font):

“Tigerfish is a computational pipeline composed of a collection of Python scripts embedded in an automated Snakemake workflow⁶¹ that chains together novel code purpose built for Tigerfish and calls to existing bioinformatic tools that are commonly used to solve problems such as parallelized sequence alignment and *k*-mer counting. Tigerfish is designed to be executed in a command line environment.”

In terms of the novelty of the use case, as the authors acknowledge it is already possible to design FISH probes for highly repetitive DNA regions with existing bioinformatics packages such as OligoMiner + PaintSHOP. The authors benchmarked the performance of this combination on human chromosome-specific alpha satellite arrays, on either the full array or consensus monomers discovered by Tandem Repeat Finder. While it is true that it takes too long to filter probes out for specificity if all probes discovered on the full array are considered, the CPU resource usage for discovering and then filtering probes from consensus monomers is quite reasonable (1 ~ 2 h for the four examples the authors showed). Comparing this to the CPU resource usage of TigerFISH in Figure 5, where ~ 90 % of repetitive DNA regions also take 1 ~ 2 CPU hours to design probes for, the resource usage of the already published OligoMiner + PaintSHOP combination seems to fall within this range as well. Therefore, based on the four chromosome-specific alpha satellite array examples the authors provided, it is hard to establish that TigerFISH indeed outperforms previous methods in computational efficiency for the purpose of designing probes for highly repetitive DNA regions.

Response: We thank the Reviewer for raising this point. While we agree that it is possible to design probes using existing packages, we feel the key point is that there is no straightforward way to evaluate their specificity with using the *in silico* simulation framework to predict the interval-specific on- and off-target binding properties of probes containing highly reiterated DNA sequences that Tigerfish introduces. As the vast majority of candidate probes designed using the full HOR arrays or their consensus have very significant predicted off-target binding activity, researchers would ultimately either need to spend significant effort validating individual probes experimentally—which would not be feasible for most targets—or to create a framework for predicting the specificity of candidate probes, which requires significant computational expertise. Tigerfish was born out of the need for this framework.

We thank the Reviewer for raising the point that starting with the consensus monomer vs. the full arrays can speed up the design process and reduce its resource usage. While this will not be possible for all targets (e.g., chr21 and chrY both returned 0 viable probes when using the consensus monomers, but 6 and 17, respectively, with the full arrays), it is a good place for researchers to start if a consensus is available. We have added the following text to the **Results** section of the manuscript to incorporate this

helpful suggestion (page 7, line 33, new text in blue font):

“Probe Design Mode is intended for instances where the genomic interval(s) a user wants to target for probe design are already known. In this case, the user must provide an additional BED-formatted file⁶⁸ that specifies the genomic coordinates for interval(s) to perform probe design against; these coordinates can refer to the full interval or, if available, the location of a consensus monomer. The third, termed “Probe Analysis Mode”, runs the pipeline starting at the specificity filtering step that comes downstream of probe design (**Fig. 1**).”